# Improved Analysis for Sign-based Methods with Momentum Updates

## Abstract

This paper presents enhanced analysis for sign-based optimization algorithms with momentum updates. Traditional sign-based methods obtain a convergence rate of $\mathcal{O}(T^{-1/4})$ under the separable smoothness assumption, but they typically require large batch sizes or assume unimodal symmetric stochastic noise. To address these limitations, we demonstrate that signSGD with momentum can achieve the same convergence rate using constant batch sizes without additional assumptions. We also establish a convergence rate under the $l_2$-smoothness condition, improving upon the result of the prior momentum-based signSGD variant by a factor of $\mathcal{O}(d^{1/2})$, where $d$ is the problem dimension. Furthermore, we explore sign-based methods with majority vote in distributed settings and show that the proposed momentum-based method yields convergence rates of $\mathcal{O}\left(d^{1/2}T^{-1/2} + dn^{-1/2}\right)$ and $\mathcal{O}\left(\max\{d^{1/4}T^{-1/4}, d^{1/10}T^{-1/5}\}\right)$, which outperform the previous results of $\mathcal{O}\left(dT^{-1/4} + dn^{-1/2}\right)$ and $\mathcal{O}\left(d^{3/8}T^{-1/8}\right)$, respectively. Numerical experiments also validate the effectiveness of the proposed methods.

## 1 Introduction

This paper investigates the stochastic optimization problem in the form

$$\min_{\mathbf{x}\in\mathbb{R}^d} f(\mathbf{x}), \tag{1}$$

where $f : \mathbb{R}^d \to \mathbb{R}$ is a smooth function. We assume that only noisy estimations of the gradient are available, denoted as $\nabla f(\mathbf{x};\xi)$, where $\xi$ is a random sample such that $\mathbb{E}[\nabla f(\mathbf{x};\xi)] = \nabla f(\mathbf{x})$.

Problem (1) has been extensively studied in the literature (Duchi et al., 2011; Kingma & Ba, 2015; Loshchilov & Hutter, 2017; Fang et al., 2018; Wang et al., 2019). One of the most widely used methods for this problem is Stochastic Gradient Descent (SGD), which updates the parameters as:

$$\mathbf{x}_{t+1} = \mathbf{x}_t - \eta\nabla f(\mathbf{x}_t;\xi_t), \tag{2}$$

where $\eta$ is the learning rate and $\xi_t$ is the random sample drawn at the $t$-th iteration. It is known that SGD achieves a convergence rate of $\mathcal{O}(T^{-1/4})$, where $T$ is the number of iterations (Ghadimi & Lan, 2013). This rate is proved to be optimal under standard assumptions (Arjevani et al., 2023).

Instead of using the stochastic gradient to update, several works (Bernstein et al., 2018; 2019; Safaryan & Richtarik, 2021) propose to update using only the sign of the stochastic gradient, i.e.,

$$\mathbf{x}_{t+1} = \mathbf{x}_t - \eta\,\mathrm{sign}\left(\nabla f(\mathbf{x}_t;\xi_t)\right), \tag{3}$$

which is particularly beneficial in distributed settings. In such scenarios, only the sign information needs to be transmitted between nodes, significantly reducing communication overhead.

Recently, several studies have investigated the convergence properties of signSGD and its variants. Bernstein et al. (2018) first prove that signSGD achieves a convergence rate of $\mathcal{O}(N^{-1/4})$ under the separable smoothness assumption, where $N$ is the number of stochastic gradient calls. However, their analysis requires a large batch size of $\mathcal{O}(\sqrt{N})$ in each iteration. Later, Bernstein et al. (2019) demonstrate that signSGD can achieve the same convergence rate with constant batch sizes, but under the additional assumption that the noise is unimodal and symmetric. To avoid such extra

Table 1: Summary of convergence rates for sign-based algorithms. Here, $T$ represents the number of stochastic gradient calls and $l_1$&$l_2$ denotes mixed $l_1$-norm and weighted $l_2$-norm. We use stochastic gradient calls rather than iteration numbers to measure convergence, in order to provide a fairer comparison across different algorithms with varying batch sizes.

| Method | Convergence | Assumptions | Measure | Additional Requirements |
|---|---|---|---|---|
| signSGD (Bernstein et al., 2018) | $\mathcal{O}\left(\frac{1}{T^{1/4}}\right)$ | | $l_1$ | Large batch size of $\mathcal{O}(\sqrt{T})$ |
| Signum (Bernstein et al., 2018) | $\tilde{\mathcal{O}}\left(\frac{1}{T^{1/4}}\right)$ | | $l_1$ | Large batch size of $\mathcal{O}(\sqrt{T})$ |
| signSGD (Bernstein et al., 2019) | $\mathcal{O}\left(\frac{1}{T^{1/4}}\right)$ | Assumptions 1, 2, 3 | $l_1$&$l_2$ | Unimodal symmetric noise |
| **Theorem 1** (this work) | $\mathcal{O}\left(\frac{1}{T^{1/4}}\right)$ | | $l_1$ | – |
| signSGD-SIM (Sun et al., 2023) | $\mathcal{O}\left(\frac{d}{T^{1/4}}\right)$ | Assumptions 1, 4, 5 | $l_1$ | – |
| **Theorem 2** (this work) | $\mathcal{O}\left(\frac{d^{1/2}}{T^{1/4}}\right)$ | | | |

assumptions, Sun et al. (2023) show that signSGD with momentum can achieve a convergence rate of $\mathcal{O}(dT^{-1/4})$ under the $l_2$-smoothness assumption. However, this dependence on $d$ is unsatisfactory, leading to high sample complexity for high-dimensional problems.

In this paper, we re-examine signSGD with momentum and establish a convergence rate of $\mathcal{O}(T^{-1/4})$ under the separable smoothness condition. Compared with previous work (Bernstein et al., 2018; 2019), our analysis does not require large batch sizes or unimodal symmetric noise. Under the $l_2$-smoothness assumption, we also derive a convergence rate of $\mathcal{O}(d^{1/2}T^{-1/4})$, improving the previous result of $\mathcal{O}(dT^{-1/4})$ (Sun et al., 2023).

For distributed sign-based methods, each node typically transmits the sign of its gradient to the server, which then sends back the sign of the aggregated gradients for update. In this context, previous literature establishes convergence rates of $\mathcal{O}\left(\frac{d}{T^{1/4}} + \frac{d}{n^{1/2}}\right)$ (Sun et al., 2023) and $\mathcal{O}\left(\frac{d^{3/8}}{T^{1/8}}\right)$ (Jin et al., 2021), where $n$ denotes the number of nodes. To improve these rates, we utilize an unbiased sign operation along with momentum updates, achieving convergence rates of $\mathcal{O}\left(\frac{d^{1/2}}{T^{1/2}} + \frac{d}{n^{1/2}}\right)$, $\mathcal{O}\left(\frac{n^{1/2}}{T} + \frac{d}{n^{1/2}}\right)$ and $\mathcal{O}\left(\max\{\frac{d^{1/4}}{T^{1/4}}, \frac{d^{1/10}}{T^{1/5}}\}\right)$, with different hyper-parameter settings and algorithm designs. In summary, this paper makes the following contributions:

- Under the separable smoothness assumption, we prove that signSGD with momentum can achieve a convergence rate of $\mathcal{O}(T^{-1/4})$ without additional assumptions. In contrast, existing analyses require either large batches or the assumption of unimodal symmetric noise.

- Under the $l_2$-smoothness assumption, we show that signSGD with momentum achieves a convergence rate of $\mathcal{O}(d^{1/2}T^{-1/4})$, improving upon the $\mathcal{O}(dT^{-1/4})$ result of the existing momentum-based signSGD method under the same conditions.

- In distributed settings, we derive convergence rates of $\mathcal{O}\left(\frac{n^{1/2}}{T} + \frac{d}{n^{1/2}}\right)$, $\mathcal{O}\left(\frac{d^{1/2}}{T^{1/2}} + \frac{d}{n^{1/2}}\right)$ and $\mathcal{O}\left(\max\left\{\frac{d^{1/4}}{T^{1/4}}, \frac{d^{1/10}}{T^{1/5}}\right\}\right)$, with the latter two substantially outperforming previous results of $\mathcal{O}\left(\frac{d}{T^{1/4}} + \frac{d}{n^{1/2}}\right)$ and $\mathcal{O}\left(\frac{d^{3/8}}{T^{1/8}}\right)$, respectively.

We compare our results with existing methods in Tables 1 and 2.

Table 2: Summary of results for sign-based algorithms in the distributed setting, where $n$ represents the number of nodes and $T$ denotes the iteration number.

| Method | Convergence | Measure |
|---|---|---|
| MV-sto-signSGD-SIM
(Sun et al., 2023) | $\mathcal{O}\left(\frac{d}{T^{1/4}} + \frac{d}{n^{1/2}}\right)$ | |
| **Theorem 3**
(this work) | $\mathcal{O}\left(\frac{d^{1/2}}{T^{1/2}} + \frac{d}{n^{1/2}}\right)$ | $l_1$ |
| **Theorem 4**
(this work) | $\mathcal{O}\left(\frac{n^{1/2}}{T} + \frac{d}{n^{1/2}}\right)$ | |
| Sto-signSGD
(Jin et al., 2021) | $\mathcal{O}\left(\frac{d^{3/8}}{T^{1/8}}\right)$ | $l_2$ |
| **Theorem 5**
(this work) | $\mathcal{O}\left(\max\left\{\frac{d^{1/4}}{T^{1/4}}, \frac{d^{1/10}}{T^{1/5}}\right\}\right)$ | |

## 2 RELATED WORK

In this section, we review the signSGD method and its variants, as well as sign-based methods with majority vote in distributed settings.

### 2.1 SIGNSGD AND ITS VARIANTS

The convergence of signSGD is first analyzed by Bernstein et al. (2018), who obtain a rate of $\mathcal{O}(N^{-1/4})$ with a large batch size of $\mathcal{O}(\sqrt{N})$, where $N$ is the number of stochastic gradient calls. They also show that the momentum version of signSGD, named Signum, achieves a convergence rate of $\mathcal{O}(N^{-1/4}\log N)$ with increasingly large batches. To avoid large batch sizes, Bernstein et al. (2019) attain the same convergence rate with a constant batch size, but rely on the strong assumption that the stochastic gradient noise is both unimodal and symmetric, which is not satisfied for many types of noise in practice.

Subsequently, Karimireddy et al. (2019) observe that signSGD with a constant batch size may not converge to optimal points for convex objectives and performs poorly compared to traditional SGD. To address this, they incorporate the compression error into the next update step and show that error feedback enhances practical performance. However, their error-feedback method needs to transmit additional information and further assumes the bounded gradient assumption, making their analyses non-standard. Rather than assuming unbiased estimation and bounded noise, Safaryan & Richtarik (2021) provide convergence guarantees under the success probability bounds assumption, which posits that the sign of the stochastic gradient matches that of the true gradient with a probability greater than $1/2$. Recently, Sun et al. (2023) analyze the momentum-based version of signSGD and achieve a convergence rate of $\mathcal{O}(dT^{-1/4})$ under standard assumptions. However, their dependence on $d$ can be further improved, as demonstrated by our analysis.

Besides, several other variants have been proposed. For instance, ZO-signSGD (Liu et al., 2019) combines zeroth-order updates with sign information, ensuring gradient-free and communication compression. Jiang et al. (2024) incorporate variance reduction with sign operation, improving the convergence to $\mathcal{O}(T^{-1/3})$ under a stronger smoothness assumption and to $\mathcal{O}(d^{1/2}m^{1/4}T^{-1/2})$ for finite-sum problems, where $m$ denotes the number of functions in the finite-sum structure. However, their $\mathcal{O}(T^{-1/3})$ result is obtained under the stronger average smoothness assumption, which requires that each stochastic sample is smooth. Additionally, their proposed method involves computing the gradient at the previous decision point, resulting in additional computational overhead.

### 2.2 SIGN-BASED METHODS WITH MAJORITY VOTE

The majority vote technique is employed to enable communication compression in distributed settings. In this framework, each node transmits only the sign of its gradient estimation to the parameter server, which then aggregates the information and sends the sign of the aggregated data

back to each node for updating. In the homogeneous setting, Bernstein et al. (2018) first demonstrate that signSGD with majority vote can achieve a convergence rate of $\mathcal{O}(T^{-1/4})$ with large batch sizes. Later, Bernstein et al. (2019) further obtain the same rate with a constant batch size when the noise is unimodal and symmetric. For more challenging heterogeneous environments, the SSDM method (Safaryan & Richtarik, 2021) achieves a convergence rate of $\mathcal{O}(d^{1/2}T^{-1/4})$ under the success probability bounds assumption. However, SSDM only guarantees 1-bit compression in one direction, since the information sent back to each node is not the sign information anymore. To address this, Stochastic-Sign SGD (Jin et al., 2021) ensures 1-bit compression in both directions and achieves a convergence rate of $\mathcal{O}(d^{3/8}T^{-1/8})$ in terms of the $l_2$-norm. Later, Sun et al. (2023) propose the MV-sto-signSGD-SIM method, attaining a convergence rate of $\mathcal{O}\left(\frac{d}{T^{1/4}} + \frac{d}{n^{1/2}}\right)$. By incorporating variance reduction techniques, Jiang et al. (2024) improve the convergence rates to $\mathcal{O}\left(\frac{d^{1/2}}{T^{1/2}} + \frac{d}{n^{1/2}}\right)$ and $\mathcal{O}(d^{1/4}T^{-1/4})$, under a stronger average smoothness assumption.

# 3 SIGNSGD WITH MOMENTUM UPDATES

In this section, we first introduce the assumptions used to analyze sign-based methods and then present our convergence guarantees for signSGD with momentum. Due to space limitations, all proofs are deferred to the Appendix.

## 3.1 ASSUMPTIONS

We outline the assumptions commonly used to derive convergence guarantees for sign-based methods (Bernstein et al., 2018; 2019).

**Assumption 1** $f_* = \inf_x f(x) > -\infty$ and $f(\mathbf{x}_1) - f_* \leq \Delta_f$ for the initial solution $\mathbf{x}_1$.

**Assumption 2** (Separable smoothness) The objective function $f$ is separable smooth if there exist non-negative constants $[L_1, L_2, \ldots, L_d]$ such that

$$f(\mathbf{y}) \leq f(\mathbf{x}) + \langle \nabla f(\mathbf{x}), \mathbf{y} - \mathbf{x} \rangle + \frac{1}{2}\sum_{i=1}^{d} L_i(\mathbf{y}_i - \mathbf{x}_i)^2.$$

**Assumption 3** (Separable bounded noise) For non-negative constants $[\sigma_1, \sigma_2, \cdots, \sigma_d]$, we have

$$\mathbb{E}_\xi\left[\left([\nabla f(\mathbf{x};\xi)]_i - [\nabla f(\mathbf{x})]_i\right)^2\right] \leq \sigma_i^2.$$

Instead of using Assumptions 2 and 3, other literature (Sun et al., 2023; Jiang et al., 2024) employs the following assumptions alternatively.

**Assumption 4** ($l_2$-smoothness) The objective function $f$ is L-smooth if
$$\|\nabla f(\mathbf{x}) - \nabla f(\mathbf{y})\| \leq L\|\mathbf{x} - \mathbf{y}\|.$$

**Assumption 5** (Bounded noise) The stochastic gradient noise is bounded such that
$$\mathbb{E}_\xi\left[\|\nabla f(\mathbf{x};\xi) - \nabla f(\mathbf{x})\|^2\right] \leq \sigma^2.$$

**Remark:** To align with different literature, we provide two distinct theorems in the next subsection, derived under Assumptions 1, 2, 3 and Assumptions 1, 4, 5 respectively.

## 3.2 THE CONVERGENCE GUARANTEES

Here, we introduce the sign-based method with momentum updates and present the corresponding convergence guarantees. The traditional signSGD method uses the sign of the stochastic gradient for updates, in the form of equation (3). In contrast to the signSGD method, we track the gradient using a momentum estimator $\mathbf{v}_t$, defined as

$$\mathbf{v}_t = (1-\beta)\mathbf{v}_{t-1} + \beta\nabla f(\mathbf{x}_t;\xi_t), \tag{4}$$

---

**Algorithm 1** Signum

---

1: **Input:** iteration number $T$, initial point $\mathbf{x}_1$
2: **for** time step $t = 1$ **to** $T$ **do**
3:     **if** $t == 1$ **then**
4:        Compute $\mathbf{v}_t = \nabla f(\mathbf{x}_t; \xi_t)$
5:     **else**
6:        Compute $\mathbf{v}_t = (1 - \beta)\mathbf{v}_{t-1} + \beta\nabla f(\mathbf{x}_t; \xi_t)$
7:     **end if**
8:     Update the decision variable: $\mathbf{x}_{t+1} = \mathbf{x}_t - \eta \operatorname{sign}(\mathbf{v}_t)$
9: **end for**
10: Select $\tau$ uniformly at random from $\{1, \ldots, T\}$
11: Return $\mathbf{x}_\tau$

---

where $\beta$ is the momentum parameter and we use $\mathbf{v}_1 = \nabla f(\mathbf{x}_1; \xi_1)$ for the first iteration. After computing the estimator $\mathbf{v}_t$, we update the decision variable using the sign of $\mathbf{v}_t$ as follows:

$$\mathbf{x}_{t+1} = \mathbf{x}_t - \eta \operatorname{sign}(\mathbf{v}_t). \tag{5}$$

The full algorithm is outlined in Algorithm 1, which is called Signum in the previous work (Bernstein et al., 2018) (also named as signSGD-SIM by Sun et al. (2023)). Our contribution lies in the improved theoretical analysis. To compare with previous signSGD studies (Bernstein et al., 2018; 2019), we first provide guarantees under the separable smoothness assumption.

**Theorem 1** *Under Assumptions 1, 2 and 3, by setting $\beta = \mathcal{O}\left(T^{-1/2}\right)$ and $\eta = \mathcal{O}\left(T^{-3/4}\right)$, Algorithm 1 ensures that*

$$\mathbb{E}\left[\|\nabla f(\mathbf{x}_\tau)\|_1\right] \leq \mathcal{O}\left(\frac{1}{T^{1/4}}\right).$$

**Remark:** The above rate implies a sample complexity of $\mathcal{O}(\epsilon^{-4})$, matching the state-of-the-art results for signSGD (Bernstein et al., 2018; 2019). However, our method does not require large batch sizes which can be as large as $\mathcal{O}(\epsilon^{-2})$ for signSGD (Bernstein et al., 2018), and avoids the unimodal symmetric noise assumption required by Bernstein et al. (2019).

Next, we also provide the theoretical guarantee under the $l_2$-smoothness assumption.

**Theorem 2** *Under Assumptions 1, 4 and 5, by setting $\beta = \mathcal{O}\left(T^{-1/2}\right)$ and $\eta = \mathcal{O}\left(d^{-1/2}T^{-3/4}\right)$, Algorithm 1 ensures that*

$$\mathbb{E}\left[\|\nabla f(\mathbf{x}_\tau)\|_1\right] \leq \mathcal{O}\left(\frac{d^{1/2}}{T^{1/4}}\right).$$

**Remark:** This rate implies a sample complexity of $\mathcal{O}(d^2\epsilon^{-4})$, an improvement over the $\mathcal{O}(d^4\epsilon^{-4})$ results of previous sign-based momentum methods (Sun et al., 2023). This improvement is especially significant when the dimension $d$ is large.

**Remark:** In Theorem 1, by using the separable smoothness and separable bounded noise assumptions (Assumptions 2 and 3), we can directly analyze under the $\ell_1$-norm and provide coordinate-wise bounds, thus avoiding the $d^{1/2}$ dependency.

**Source of Theoretical Improvement:** In the previous work (Bernstein et al., 2018), to bound the term $\sum_i |[\nabla f(\mathbf{x}_t)]_i| \cdot \mathbb{P}\left(\operatorname{sign}([\nabla f(\mathbf{x}_t)]_i) \neq \operatorname{sign}([\mathbf{v}_t]_i)\right)$ appeared in the analysis, they apply $\mathbb{P}\left(\operatorname{sign}([\nabla f(\mathbf{x}_t)]_i) \neq \operatorname{sign}(\nabla f(\mathbf{x}_t; \xi_t)_i)\right) \leq \frac{\sigma_i}{\sqrt{n_i}|[\nabla f(\mathbf{x}_t)]_i|}$, which inevitably requires *huge batch sizes* $n_i$ to ensure convergence. Later work (Bernstein et al., 2019) assumes *unimodal symmetric noise* to deal with $\mathbb{P}\left(\operatorname{sign}([\nabla f(\mathbf{x}_t)]_i) \neq \operatorname{sign}([\mathbf{v}_t]_i)\right)$. While in our analysis, we find that standard assumption is already adequate and we use $\sum_i |[\nabla f(\mathbf{x}_t)]_i| \cdot \mathbb{P}\left(\operatorname{sign}([\nabla f(\mathbf{x}_t)]_i) \neq \operatorname{sign}([\mathbf{v}_t]_i)\right) \leq \sum_i |[\nabla f(\mathbf{x}_t)]_i - [\mathbf{v}_t]_i| \leq \|\nabla f(\mathbf{x}_t) - \mathbf{v}_t\|_1$ in the analysis. Since we further provide a tighter bound for the estimation error $\|\nabla f(\mathbf{x}_t) - \mathbf{v}_t\|_1$ compared to Sun et al. (2023), we achieve the state-of-the-art convergence rate without relying on additional assumptions.

### 3.3 SHARPNESS OF THE OBTAINED RATES

The convergence lower bound for stochastic optimization is $\Omega(T^{-1/4})$ in the $l_2$-norm (Arjevani et al., 2023). Since we know that $\|z\|_1 \geq \|z\|_2$, this lower bound also implies that $\mathbb{E}[\|\nabla f(\mathbf{x}_\tau)\|_1] \geq \mathbb{E}[\|\nabla f(\mathbf{x}_\tau)\|_2] \geq \Omega(T^{-1/4})$, indicating that our result is optimal with respect to $T$.

Regarding the $d^{1/2}$ factor in the convergence rate, several pieces of evidence suggest that this factor is inherent for $l_1$-norm convergence under the standard $l_2$-smoothness assumption:

- Jiang et al. (2025) establish an $\Omega\left(\sqrt{\frac{d\|L\|_\infty}{T}} + \frac{d^{1/4}\left(\sum_{i=1}^d \sigma_i \sqrt{L_i}\right)^{1/2}}{T^{1/4}}\right)$ lower bound for SGD when measured with the $l_1$-norm. Suppose that $\{\sigma_i\}$ and $\{L_i\}$ have the same value across coordinates such that $\sigma_i = \sigma/\sqrt{d}, L_i = L$, the lower bound becomes $\Omega\left(\frac{d^{1/2}}{T^{1/4}}\right)$, which confirms the $\sqrt{d}$ factor is required in the $l_1$-norm setting.
- Prior works (Bernstein et al., 2018; Dong et al., 2024) have already conducted extensive experiments on various vision and language tasks, and find that the ratio of the gradient norm $r = \|\nabla f(x)\|_1 / \|\nabla f(x)\|_2$ always stay close to the level of $\Theta(\sqrt{d})$, supporting the presence of the $\sqrt{d}$ factor in the $l_1$ measure from the empirical sense.
- Existing rates for sign-based methods under the $l_1$-norm and $l_2$-smoothness assumption also include the $\sqrt{d}$ dependency, or even worse (Jin et al., 2021; Sun et al., 2023). Our Theorem 2 already improves the $d$-dependency from Sun et al. (2023) under the same assumptions.

## 4 MAJORITY VOTE SIGNSGD WITH MOMENTUM UPDATES

We first present the problem formulation and the assumptions used. Then, we introduce the proposed method and establish the convergence guarantees.

### 4.1 PROBLEM FORMULATION AND ASSUMPTIONS

Sign-based methods are highly communication-efficient in distributed settings, as they only require 1-bit sign information for updates. Previous literature (Bernstein et al., 2018; 2019; Jin et al., 2021; Sun et al., 2023) has analyzed sign-based methods with majority vote in distributed environments. To begin with, consider the following distributed learning problem:

$$\min_{\mathbf{x} \in \mathbb{R}^d} f(\mathbf{x}) := \frac{1}{n} \sum_{j=1}^n f_j(\mathbf{x}), \quad f_j(\mathbf{x}) := \mathbb{E}_{\xi^j \sim \mathcal{D}_j}\left[f_j(\mathbf{x}; \xi^j)\right], \tag{6}$$

where $\mathcal{D}_j$ represents the data distribution on node $j$, and $f_j(\mathbf{x})$ is the corresponding loss function.

Early studies (Bernstein et al., 2018; 2019) focus on homogeneous settings, where $\mathcal{D}_j$ and $f_j$ are identical across nodes. For the more difficult heterogeneous setting, Jin et al. (2021) derive a convergence rate of $\mathcal{O}\left(d^{3/8}T^{-1/8}\right)$ and Sun et al. (2023) achieve the rate of $\mathcal{O}\left(\frac{d}{T^{1/4}} + \frac{d}{n^{1/2}}\right)$. However, these rates can still be improved based on our analysis.

Next, we introduce the assumptions required in this section, which are standard and commonly used in previous works (Jin et al., 2021; Sun et al., 2023).

**Assumption 6** *(Smoothness on node $j$) For each node $j \in [n]$, we suppose*
$$\|\nabla f_j(\mathbf{x}) - \nabla f_j(\mathbf{y})\| \leq L \|\mathbf{x} - \mathbf{y}\|.$$

**Assumption 7** *(Bounded noise on node $j$) For each node $j \in [n]$, we have*
$$\mathbb{E}_\xi\left[\|\nabla f_j(\mathbf{x}; \xi) - \nabla f_j(\mathbf{x})\|^2\right] \leq \sigma^2.$$

**Assumption 8** *(Bounded gradients) For each node $j \in [n]$, we assume $\sup_{\mathbf{x}} \|\nabla f_j(\mathbf{x}; \xi)\|_\infty \leq G$.*

**Remark:** The bounded gradients assumption is standard and widely employed for sign-based optimization in heterogeneous settings (Jin et al., 2021; Sun et al., 2023; Tang et al., 2024). Also note that our Assumption 8 is strictly weaker than the one used by Sun et al. (2023), which requires bounded gradients in the $l_2$-norm, i.e., $\sup_{\mathbf{x}} \|\nabla f_j(\mathbf{x}; \xi)\|_2 \leq G$.

## 4.2 THE PROPOSED METHOD

In this subsection, we introduce the proposed method for the heterogeneous distributed environments and aim to obtain improved convergence rates without additional strong assumptions. For distributed settings, the most straightforward approach is to apply the sign operation twice:

$$\mathbf{x}_{t+1} = \mathbf{x}_t - \eta \operatorname{sign}\left(\frac{1}{n}\sum_{j=1}^{n}\operatorname{sign}(\mathbf{v}_t^j)\right), \tag{7}$$

where $\mathbf{v}_t^j$ is the gradient estimator at node $j$. In this formulation, each node transmits the sign of its gradient estimate $\operatorname{sign}(\mathbf{v}_t^j)$ to the server. The server then aggregates these sign values and broadcasts back the sign of the resulting information $\operatorname{sign}\left(\frac{1}{n}\sum_{j=1}^{n}\operatorname{sign}(\mathbf{v}_t^j)\right)$ to each node for updating. This approach ensures 1-bit communication in both directions. However, the sign operation introduces bias in the estimation, and applying it twice can significantly amplify this bias. To mitigate this, we introduce an unbiased sign operation (Sun et al., 2023) as stated below.

**Definition 1** *For any vector $\mathbf{v}$ with $\|\mathbf{v}\|_\infty \leq R$, the function $S_R(\mathbf{v})$ is defined component-wise by:*

$$[S_R(\mathbf{v})]_k = \begin{cases} 1, & \text{with probability } \frac{R+[\mathbf{v}]_k}{2R}, \\ -1, & \text{with probability } \frac{R-[\mathbf{v}]_k}{2R}. \end{cases} \tag{8}$$

**Remark:** This operation provides an unbiased estimate of $\mathbf{v}/R$, since $\mathbb{E}[S_R(\mathbf{v})] = \mathbf{v}/R$.

We can now introduce our majority vote signSGD with momentum updates. First, we use the momentum gradient estimator at each node $j$ as follows:

$$\mathbf{v}_t^j = (1-\beta)\mathbf{v}_{t-1}^j + \beta\nabla f_j(\mathbf{x}_t;\xi_t^j), \tag{9}$$

where $\beta$ is the momentum parameter. Next, by communicating the gradient estimators with the unbiased sign operation $S_G(\cdot)$, we update the decision variable as follows:

$$\mathbf{x}_{t+1} = \mathbf{x}_t - \eta \operatorname{Sign}\left(\frac{1}{n}\sum_{j=1}^{n}S_G(\mathbf{v}_t^j)\right). \tag{10}$$

After applying $S_G(\cdot)$, the output is a sign vector, which can be efficiently transmitted between nodes. The complete algorithm is described in Algorithm 2 (v1), called Majority Vote SignSGD with Momentum (MVSM). For $t = 1$, we initialize $\mathbf{v}_1^j = \nabla f_j(\mathbf{x}_1;\xi_1^j)$. MVSM-v1 is identical to MV-sto-signSGD-SIM (with $\alpha = 0$) from Sun et al. (2023). However, our analysis yields stronger convergence guarantees as stated below.

**Theorem 3** *Under Assumptions 1, 6, 7 and 8, by setting that $\beta = \frac{1}{2}$ and $\eta = \mathcal{O}\left(\frac{1}{T^{1/2}d^{1/2}}\right)$, our MVSM (v1) method ensures the following convergence:*

$$\mathbb{E}\left[\|\nabla f(\mathbf{x}_\tau)\|_1\right] \leq \mathcal{O}\left(\frac{d^{1/2}}{T^{1/2}} + \frac{d}{n^{1/2}}\right).$$

**Remark:** Our rate is superior to the previous result of $\mathcal{O}\left(\frac{d}{T^{1/4}} + \frac{d}{n^{1/2}}\right)$, indicating our significant improvement over prior work (Sun et al., 2023) in both $d$ and $T$ dependencies.

By adjusting the learning rate, we can also obtain the following convergence guarantee.

**Theorem 4** *Under Assumptions 1, 6, 7 and 8, by setting $\beta = \frac{1}{2}$ and $\eta = \mathcal{O}(n^{-1/2})$, our MVSM (v1) method ensures:*

$$\mathbb{E}\left[\|\nabla f(\mathbf{x}_\tau)\|_1\right] \leq \mathcal{O}\left(\frac{n^{1/2}}{T} + \frac{d}{n^{1/2}}\right).$$

---

**Algorithm 2** Majority vote signSGD with momentum (MVSM)

---

1: **Input:** iteration number $T$, initial point $\mathbf{x}_1$
2: **for** time step $t = 1$ **to** $T$ **do**
3:     **On** node $j \in \{1, 2, \cdots, n\}$:
4:         Compute $\mathbf{v}_t^j = (1 - \beta)\mathbf{v}_{t-1}^j + \beta \nabla f_j(\mathbf{x}_t; \xi_t^j)$
5:         Send $\mathrm{S}_G(\mathbf{v}_t^j)$ to the parameter server
6:     **On** parameter server:
7:         (v1) Send $\mathbf{v}_t = \mathrm{sign}\left(\frac{1}{n}\sum_{j=1}^n \mathrm{S}_G\left(\mathbf{v}_t^j\right)\right)$ to all nodes
8:         (v2) Send $\mathbf{v}_t = \mathrm{S}_1\left(\frac{1}{n}\sum_{j=1}^n \mathrm{S}_G\left(\mathbf{v}_t^j\right)\right)$ to all nodes
9:     **On** node $j \in \{1, 2, \cdots, n\}$:
10:       Update the decision variable $\mathbf{x}_{t+1} = \mathbf{x}_t - \eta \mathbf{v}_t$
11: **end for**
12: Select $\tau$ uniformly at random from $\{1, \ldots, T\}$
13: Return $\mathbf{x}_\tau$

---

**Remark:** This rate improves Theorem 3 when $T \geq \frac{n}{d}$, which is easily satisfied when $d$ is large.

**Source of Theoretical Improvement:** The improvement for Algorithm 2 (v1) lies in deriving a tighter error bound for the gradient estimator, i.e., $\epsilon_t = \left\|\nabla f(\mathbf{x}_t) - \frac{1}{n}\sum_{j=1}^n \mathbf{v}_t^j\right\|_2^2$. By carefully analyzing the aggregated estimator $\frac{1}{n}\sum_{j=1}^n \mathbf{v}_t^j$, we obtain the recurrence: $\epsilon_{t+1} = (1 - \beta)\epsilon_t + \frac{\sigma^2 \beta^2}{n} + \frac{2L^2 \eta^2 d}{\beta}$, which allows fast decay of $\epsilon_t$ with appropriate choices of $\beta$ and $\eta$.

Although the above theorems achieve better convergence rates than previous methods, they do not converge to zero as $T$ increases. To address this issue, we replace the sign operation in the server with the unbiased sign operation $\mathrm{S}_1(\cdot)$ as defined in equation (8) with $R = 1$. The revised formulation for the update is:

$$\mathbf{v}_t = \mathrm{S}_1\left(\frac{1}{n}\sum_{j=1}^n \mathrm{S}_G\left(\mathbf{v}_t^j\right)\right). \tag{11}$$

The corresponding algorithm is presented in Algorithm 2 (v2), with the only modification in Step 8. We now present the convergence guarantee for this modified approach.

**Theorem 5** *Under Assumptions 1, 6, 7 and 8, by setting that $\eta = \mathcal{O}\left(\min\left\{\frac{1}{T^{1/2}d^{1/2}}, \frac{1}{T^{3/5}d^{1/5}}\right\}\right)$ and $\beta = \mathcal{O}\left(\eta^{2/3}d^{1/3}\right)$, the MVSM (v2) method ensures the following convergence:*

$$\mathbb{E}\left[\|\nabla f(\mathbf{x}_\tau)\|_2\right] \leq \mathcal{O}\left(\max\left\{\frac{d^{1/4}}{T^{1/4}}, \frac{d^{1/10}}{T^{1/5}}\right\}\right).$$

**Remark:** This convergence rate approaches zero as $T \to \infty$, and significantly improves upon the previous result of $\mathcal{O}\left(\frac{d^{3/8}}{T^{1/8}}\right)$ (Jin et al., 2021), in terms of both $T$ and $d$.

**Source of Theoretical Improvement:** The improved rate stems from the unbiased estimation of the full gradient, allowing us to use $\mathbb{E}\left[S_1\left(\frac{1}{n}\sum_{j=1}^n S_G(\mathbf{v}_t^j)\right)\right] = \frac{1}{nG}\sum_{j=1}^n \mathbf{v}_t^j$ in the analysis. In contrast, prior works (Jin et al., 2021; Sun et al., 2023) use biased sign operators on the server, which leads to looser bounds and higher complexities.

## 5 EXPERIMENTS

In this section, we evaluate the performance of our methods through numerical experiments. We first assess the Signum algorithm on the image classification task in a centralized setting, and then test the proposed MVSM method in the distributed learning environment. Finally, we experiment on the fine-tuning task for large language models. All experiments are conducted on NVIDIA GeForce

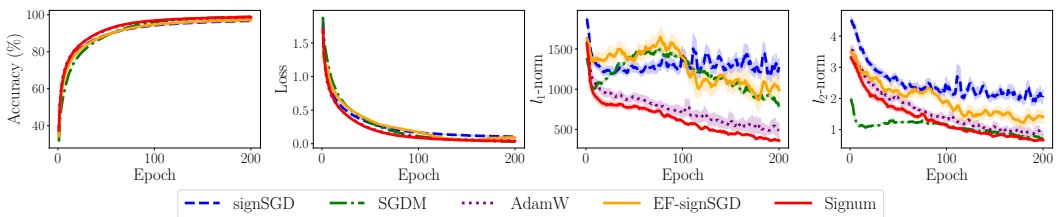

Figure 1: Results for CIFAR-10 dataset in the centralized environment.

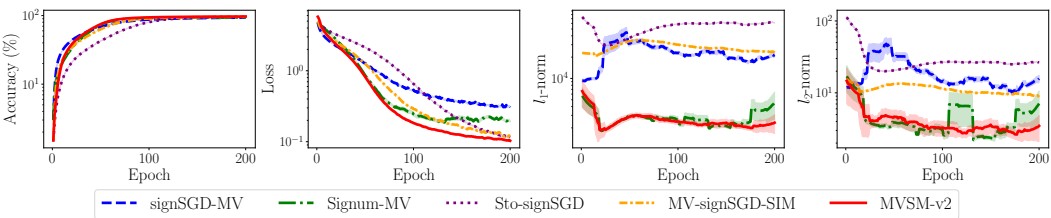

Figure 2: Results for CIFAR-100 dataset in the distributed environment.

RTX 3090 GPUs, and results are averaged over 10 runs, with shaded regions representing the standard deviation.

## 5.1 IMAGE CLASSIFICATION IN THE CENTRALIZED ENVIRONMENT

We validate the effectiveness of sign-based methods on the image classification task. Specifically, we train a ResNet-18 model (He et al., 2016) on the CIFAR-10 dataset (Krizhevsky, 2009) and compare our method against signSGD (Bernstein et al., 2018), SGDM (Sutskever et al., 2013), AdamW (Kingma & Ba, 2015; Loshchilov & Hutter, 2019), and signSGD with error feedback (i.e., EF-signSGD) (Karimireddy et al., 2019). For SGDM and AdamW, we use the official PyTorch implementations (Paszke et al., 2019). For each optimizer method, hyperparameters are determined through grid search. Specifically, the momentum parameter $\beta$ is selected from the set $\{0.9, 0.5, 0.1, 0.01\}$, and the learning rate $\eta$ is chosen from the set $\{0.5, 0.25, 0.1, 0.05, 0.025, 0.01\} \times 10^{-2}$.

Figure 1 reports the training loss, accuracy, and the $l_1$- and $l_2$-norms of the gradients. In terms of loss and accuracy, the Signum method converges fastest even among the algorithms that update with full gradient information. Additionally, the Signum method results in the most rapid reduction of both $l_1$ and $l_2$ gradient norms. These findings are consistent with our theoretical results, further highlighting the effectiveness of momentum-based sign methods in accelerating convergence and improving optimization efficiency.

## 5.2 IMAGE CLASSIFICATION IN THE DISTRIBUTED ENVIRONMENT

Next, we evaluate our method in the distributed setting. We train a ResNet-50 model (He et al., 2016) on the CIFAR-100 dataset (Krizhevsky, 2009) across 8 nodes. We compare our newly proposed MVSM-v2 method against signSGD (with Majority Vote) (Bernstein et al., 2018), Signum (with Majority Vote) (Bernstein et al., 2019), Sto-signSGD (Jin et al., 2021), and MV-signSGD-SIM (Sun et al., 2023). Also note that the MV-signSGD algorithm is identical to our MVSM-v1 method. The hyperparameters are searched in the same way as in Section 5.1.

Figure 2 presents the training loss, accuracy, and the $l_1$- and $l_2$-norms of the gradients. Our MVSM-v2 algorithm achieves the lowest loss and highest accuracy, while also exhibiting sparser gradients compared to other methods. In contrast, sign-based optimizers that do not incorporate momentum updates—specifically, signSGD-MV and Sto-signSGD—exhibit poor performance and produce significantly larger gradients. These results further underscore the advantage of integrating momentum into sign-based optimization methods.

Table 4: Training losses of finetuning GPT-2 and Qwen3 on the Alpaca dataset.

| Method | SGDM | signSGD | EF-signSGD | AdamW | Signum |
|--------|------|---------|------------|-------|--------|
| **GPT-2** | 2.509±.005 | 2.236±.004 | 2.267±.003 | 2.183±.001 | **2.176±.002** |
| **Qwen3** | 1.677±.006 | 1.610±.003 | 1.609±.005 | 1.592±.002 | **1.592±.001** |

### 5.3 Instruction Following Fine-tuning for Large Language Models

Finally, we conduct experiments on fine-tuning LLMs to evaluate the high-dimensional and practical usage of sign-based optimizers. Specifically, we compare signSGD, SGDM, AdamW, and EF-signSGD with our Signum method on the GPT-2 (Radford et al., 2019) and Qwen3-0.6B (Yang et al., 2025) models. These optimization algorithms are evaluated on the Alpaca dataset (Taori et al., 2023), which consists of 52000 instruction-following question-answer pairs. Throughout the experiments, we follow a similar setup as Liu et al. (2025c) and set the hyperparameters listed in Table 3. All other settings remain at their

Table 3: Hyperparameter configurations.

| Hyperparameter | Value |
|----------------|-------|
| weight decay | 0.1 |
| batch size | 256 |
| max sequence length | 512 |
| gradient accumulation steps | 1 |
| epochs | 1.0 |
| learning rate schedule | cosine decay |

`transformers==4.52.4` defaults. To cut memory usage and speed up training, we leverage the LMFlow toolbox (Diao et al., 2024). For the remaining hyperparameters, we either adopt the values from the original paper or perform a grid search. For AdamW, we keep $\beta_1 = 0.9$, $\beta_2 = 0.95$, the de facto standard for training LLMs such as LLaMA (Touvron et al., 2023) and AMD-Llama-135M, whose model size closely matches ours. For every other optimizer that maintains a momentum state, we search $\beta_1 \in \{0.99, 0.95, 0.9, 0.75, 0.5, 0\}$ (represents $1-\beta$ in Algorithm 1). All methods explore learning rates in $\{1, 5\} \times \{1e1, 1e0, 1e-1, 1e-2, 1e-3, 1e-4, 1e-5, 1e-6\}$.

The experimental results are listed in Table 4, which shows that our method yields lower training loss compared to other baselines. We notice that Signum has a similar performance to AdamW, which is consistent with the empirical observations by Kunstner et al. (2023); Chen et al. (2023). These findings highlight the practical effect of sign-based methods for training complex, high-dimensional models. The optimal hyperparameters for each algorithm can be found in Appendix E.1. We also conduct hyperparameter sensitivity analysis in Appendix E.2, and empirical validation of Assumption 8 in Appendix E.3.

## 6 Conclusion

In this paper, we demonstrate that signSGD with momentum update can achieve a convergence rate of $\mathcal{O}(T^{-1/4})$ without requiring large batch sizes or assuming unimodal symmetric noise. When analyzed under the $l_2$-smoothness assumption, our method achieves a convergence rate of $\mathcal{O}(d^{1/2}T^{-1/4})$, which improves upon the previous rate of $\mathcal{O}(dT^{-1/4})$. In distributed settings, we establish convergence rates of $\mathcal{O}\left(\frac{d^{1/2}}{T^{1/2}} + \frac{d}{n^{1/2}}\right)$ and $\mathcal{O}\left(\max\left\{\frac{d^{1/4}}{T^{1/4}}, \frac{d^{1/10}}{T^{1/5}}\right\}\right)$, which significantly outperform prior results of $\mathcal{O}\left(\frac{d}{T^{1/4}} + \frac{d}{n^{1/2}}\right)$ and $\mathcal{O}\left(\frac{d^{3/8}}{T^{1/8}}\right)$. Finally, numerical experiments in different learning environments also validate the effectiveness of the proposed method.

Reproducibility statement

We provide clear explanations of all assumptions and include complete proofs of our theoretical claims in the appendix. For the experimental results, we specify the dataset, baseline methods, and hyperparameter choices.

The use of LLMs

We used large language models (LLMs) solely for minor language polishing of the manuscript. The LLMs did not contribute to research ideation, algorithm design, theoretical analysis, or experimental work. Their role was strictly limited to assisting with improving readability and grammar.

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

APPENDIX

## A    PROOF OF THEOREM 1

According to Assumption 2 and considering the update $\mathbf{x}_{t+1} - \mathbf{x}_t = -\eta \operatorname{Sign}(\mathbf{v}_t)$, we know that

$$f(\mathbf{x}_{t+1}) - f(\mathbf{x}_t) \leq \langle \nabla f(\mathbf{x}_t), \mathbf{x}_{t+1} - \mathbf{x}_t \rangle + \frac{1}{2} \sum_{i=1}^{d} L_i([\mathbf{x}_{t+1}]_i - [\mathbf{x}_t]_i)^2$$

$$\leq - \langle \nabla f(\mathbf{x}_t), \eta \operatorname{Sign}(\mathbf{v}_t) \rangle + \frac{\eta^2}{2} \sum_{i=1}^{d} L_i([\operatorname{Sign}(\mathbf{v}_t)]_i)^2$$

$$\leq - \langle \nabla f(\mathbf{x}_t), \eta \operatorname{Sign}(\nabla f(\mathbf{x}_t)) \rangle$$

$$+ \eta \langle \nabla f(\mathbf{x}_t), \operatorname{Sign}(\nabla f(\mathbf{x}_t)) - \operatorname{Sign}(\mathbf{v}_t) \rangle + \frac{\eta^2}{2} \sum_{i=1}^{d} L_i$$

$$\leq - \eta \|\nabla f(\mathbf{x}_t)\|_1 + 2\eta \|\nabla f(\mathbf{x}_t) - \mathbf{v}_t\|_1 + \frac{\eta^2}{2} \sum_{i=1}^{d} L_i,$$

where the last inequality is due to

$$\langle \nabla f(\mathbf{x}_t), \operatorname{Sign}(\nabla f(\mathbf{x}_t)) - \operatorname{Sign}(\mathbf{v}_t) \rangle$$

$$= \sum_{i=1}^{d} [\nabla f(\mathbf{x}_t)]_i \cdot (\operatorname{Sign}[\nabla f(\mathbf{x}_t)]_i - \operatorname{Sign}[\mathbf{v}_t]_i)$$

$$\leq \sum_{i=1}^{d} 2[\nabla f(\mathbf{x}_t)]_i \cdot \mathbb{I}(\operatorname{Sign}([\nabla f(\mathbf{x}_t)]_i) \neq \operatorname{Sign}[\mathbf{v}_t]_i)$$

$$\leq \sum_{i=1}^{d} 2|[\nabla f(\mathbf{x}_t)]_i - [\mathbf{v}_t]_i| \cdot \mathbb{I}(\operatorname{Sign}[\nabla f(\mathbf{x}_t)]_i \neq \operatorname{Sign}[\mathbf{v}_t]_i)$$

$$\leq \sum_{i=1}^{d} 2|[\nabla f(\mathbf{x}_t)]_i - [\mathbf{v}_t]_i| = 2\|\nabla f(\mathbf{x}_t) - \mathbf{v}_t\|_1.$$

Rearranging the obtained relation and summing up yields

$$\mathbb{E}\left[\frac{1}{T}\sum_{t=1}^{T}\|\nabla f(\mathbf{x}_t)\|_1\right] \leq \frac{\Delta_f}{\eta T} + 2\mathbb{E}\left[\frac{1}{T}\sum_{t=1}^{T}\|\nabla f(\mathbf{x}_t) - \mathbf{v}_t\|_1\right] + \frac{\eta}{2}\sum_{i=1}^{d}L_i, \qquad (12)$$

where we define $\Delta_f = f(\mathbf{x}_1) - f_*$.

Next, we proceed to bound the error term $\mathbb{E}\left[\frac{1}{T}\sum_{t=1}^{T}\|\nabla f(\mathbf{x}_t) - \mathbf{v}_t\|_1\right]$. For convenience, we define the following notations:

$$\boldsymbol{\epsilon}_t := \mathbf{v}_t - \nabla f(\mathbf{x}_t), \quad \mathbf{n}_t := \nabla f(\mathbf{x}_t; \xi_t) - \nabla f(\mathbf{x}_t), \quad \mathbf{s}_t := \nabla f(\mathbf{x}_{t-1}) - \nabla f(\mathbf{x}_t).$$

By definition, we have

$$\boldsymbol{\epsilon}_t = \mathbf{v}_t - \nabla f(\mathbf{x}_t) = (1-\beta)\mathbf{v}_{t-1} + \beta \nabla f(\mathbf{x}_t; \xi_t) - \nabla f(\mathbf{x}_t)$$

$$= (1-\beta)(\mathbf{v}_{t-1} - \nabla f(\mathbf{x}_{t-1})) + (1-\beta)(\nabla f(\mathbf{x}_{t-1}) - \nabla f(\mathbf{x}_t)) + \beta(\nabla f(\mathbf{x}_t; \xi_t) - \nabla f(\mathbf{x}_t))$$

$$= (1-\beta)\boldsymbol{\epsilon}_{t-1} + (1-\beta)\mathbf{s}_t + \beta\mathbf{n}_t.$$

Performing this recursively yields

$$\boldsymbol{\epsilon}_t = (1-\beta)^{t-1}\mathbf{n}_1 + \beta \sum_{k=2}^{t}(1-\beta)^{t-k}\mathbf{n}_k + \sum_{k=2}^{t}(1-\beta)^{t-k+1}\mathbf{s}_k,$$

where we use the fact that $\boldsymbol{\epsilon}_1 = \mathbf{v}_1 - \nabla f(\mathbf{x}_1) = \nabla f(\mathbf{x}_1; \xi_1) - \nabla f(\mathbf{x}_1) = \mathbf{n}_1$. We bound $\boldsymbol{\epsilon}_t$ via two terms $\mathtt{A}_t$ and $\mathtt{B}_t$ as follows:

$$\mathbb{E}\left[\|\boldsymbol{\epsilon}_t\|_1\right] \leq \underbrace{\mathbb{E}\left[\left\|(1-\beta)^{t-1}\mathbf{n}_1 + \beta\sum_{k=2}^{t}(1-\beta)^{t-k}\mathbf{n}_k\right\|_1\right]}_{\mathtt{A}_t} + \underbrace{\mathbb{E}\left[\left\|\sum_{k=2}^{t}(1-\beta)^{t-k+1}\mathbf{s}_k\right\|_1\right]}_{\mathtt{B}_t}$$

Firstly, we cope with $\mathtt{A}_t$ following the similar procedure as in Liu et al. (2025c, Lemma E.2). We denote the $i$-th element of the vector $\mathbf{n}_t$ by $\mathbf{n}_{t,i}$. By the Cauchy–Schwarz inequality, for any $\lambda_1, \cdots, \lambda_d > 0$, it holds that

$$\mathbb{E}\left[\left\|(1-\beta)^{t-1}\mathbf{n}_1 + \beta\sum_{k=2}^{t}(1-\beta)^{t-k}\mathbf{n}_k\right\|_1^2\right]$$

$$\leq \left(\sum_{i=1}^{d}\lambda_i\right)\sum_{i=1}^{d}\frac{1}{\lambda_i}\mathbb{E}\left[(1-\beta)^{t-1}\mathbf{n}_{1,i} + \beta\sum_{k=2}^{t}(1-\beta)^{t-k}\mathbf{n}_{k,i}\right]^2$$

$$= \left(\sum_{i=1}^{d}\lambda_i\right)\sum_{i=1}^{d}\frac{1}{\lambda_i}\left((1-\beta)^{2t-2}\mathbb{E}\left[\mathbf{n}_{1,i}^2\right] + \beta^2\sum_{k=2}^{t}(1-\beta)^{2(t-k)}\mathbb{E}\left[\mathbf{n}_{k,i}^2\right]\right)$$

$$\leq \left(\sum_{i=1}^{d}\lambda_i\right)\sum_{i=1}^{d}\frac{1}{\lambda_i}\left((1-\beta)^{2t-2}\sigma_i^2 + \beta^2\sum_{k=2}^{t}(1-\beta)^{2(t-k)}\sigma_i^2\right)$$

$$\leq \left(\sum_{i=1}^{d}\lambda_i\right)\sum_{i=1}^{d}\frac{\sigma_i^2}{\lambda_i}\left((1-\beta)^{2t-2} + \frac{\beta}{2-\beta}\right),$$

where the equality is due to $\mathbb{E}\left[\mathbf{n}_{s,i}\cdot\mathbf{n}_{t,i}\right] = 0, \forall s < t \in [T], \forall i \in [d]$; the second inequality is due to Assumption 3. Denoting by $\boldsymbol{\sigma} = [\sigma_1, \cdots, \sigma_d]^\top$ and setting $\lambda_i = \sigma_i$, we obtain

$$\mathtt{A}_t \leq \sqrt{\mathbb{E}\left[\left\|(1-\beta)^{t-1}\mathbf{n}_1 + \beta\sum_{k=2}^{t}(1-\beta)^{t-k}\mathbf{n}_k\right\|_1^2\right]}$$

$$\leq \sqrt{\left((1-\beta)^{2t-2} + \frac{\beta}{2-\beta}\right)\|\boldsymbol{\sigma}\|_1^2} \leq \left((1-\beta)^{t-1} + \sqrt{\frac{\beta}{2-\beta}}\right)\|\boldsymbol{\sigma}\|_1,$$

where we make use of $\mathbb{E}^2[X] \leq \mathbb{E}[X^2]$ and $\sqrt{a+b} \leq \sqrt{a} + \sqrt{b}, \forall a, b \geq 0$.

Secondly, we cope with $\mathtt{B}_t$ as the following:

$$\mathtt{B}_t \leq \sum_{k=2}^{t}(1-\beta)^{t-k+1}\mathbb{E}\left[\|\mathbf{s}_k\|_1\right] \leq 2\eta\|\vec{L}\|_1\sum_{k=2}^{t}(1-\beta)^{t-k+1} \leq \frac{2(1-\beta)\eta\|\vec{L}\|_1}{\beta},$$

where the second inequality uses

$$\mathbb{E}\left[\|\mathbf{s}_k\|_1\right] = \|\nabla f(\mathbf{x}_{t-1}) - \nabla f(\mathbf{x}_t)\|_1 = \|\nabla f(\mathbf{x}_t + \eta\,\mathrm{Sign}\,(\mathbf{v}_{t-1})) - \nabla f(\mathbf{x}_t)\|_1 \leq 2\eta\|\vec{L}\|_1,$$

which is due to the following lemma.

**Lemma 1** *(Lemma F.3. in Bernstein et al. (2018)) Under Assumption 2, for any sign vector $\mathbf{s} \in \{-1, 1\}^d$, any $\mathbf{x} \in \mathbb{R}^d$ and any $\eta$*

$$\|\nabla f(\mathbf{x} + \eta s) - \nabla f(\mathbf{x})\|_1 \leq 2\eta\|\vec{L}\|_1.$$

Now it suffices to combine the bounds for $\mathtt{A}_t, \mathtt{B}_t$:

$$\frac{1}{T}\sum_{t=1}^{T}\mathbb{E}\left[\|\boldsymbol{\epsilon}_t\|_1\right] \leq \frac{1}{T}\sum_{t=1}^{T}(\mathtt{A}_t + \mathtt{B}_t) \leq \left(\frac{1}{T\beta} + \sqrt{\frac{\beta}{2-\beta}}\right)\|\boldsymbol{\sigma}\|_1 + \frac{2(1-\beta)\eta\|\vec{L}\|_1}{\beta},$$

where we make use of $\sum_{t=1}^{T}(1-\beta)^{t-1} \leq 1/\beta$. Plugging this relation into equation 12 yields

$$\mathbb{E}\left[\frac{1}{T}\sum_{t=1}^{T}\|\nabla f(\mathbf{x}_t)\|_1\right] \leq \frac{\Delta_f}{\eta T} + \frac{\eta\|\vec{L}\|_1}{2} + 2\|\boldsymbol{\sigma}\|_1\left(\frac{1}{T\beta} + \sqrt{\frac{\beta}{2-\beta}}\right) + \frac{4(1-\beta)\eta\|\vec{L}\|_1}{\beta}$$

Setting $\eta = \sqrt{\frac{\Delta_f}{\|\vec{L}\|_1}} \cdot T^{-3/4}$, $\beta = \frac{1}{\sqrt{T}}$, we obtain

$$\mathbb{E}\left[\|\nabla f(\mathbf{x}_\tau)\|_1\right] \leq \sqrt{\|\vec{L}\|_1\Delta_f}\left(\frac{5}{T^{1/4}} + \frac{1}{2T^{3/4}}\right) + 2\|\boldsymbol{\sigma}\|_1\left(\frac{1}{T^{1/4}} + \frac{1}{\sqrt{T}}\right) = \mathcal{O}\left(\frac{1}{T^{1/4}}\right).$$

## B  PROOF OF THEOREM 2

Firstly, due to the $l_2$-smoothness assumption (Assumption 4), we have that

$$f(\mathbf{x}_{t+1})$$
$$\leq f(\mathbf{x}_t) + \langle\nabla f(\mathbf{x}_t), \mathbf{x}_{t+1} - \mathbf{x}_t\rangle + \frac{L}{2}\|\mathbf{x}_{t+1} - \mathbf{x}_t\|^2$$
$$= f(\mathbf{x}_t) - \eta\langle\nabla f(\mathbf{x}_t), \mathrm{Sign}(\mathbf{v}_t)\rangle + \frac{\eta^2 L}{2}\|\mathrm{Sign}(\mathbf{v}_t)\|^2$$
$$\leq f(\mathbf{x}_t) + \eta\langle\nabla f(\mathbf{x}_t), \mathrm{Sign}(\nabla f(\mathbf{x}_t)) - \mathrm{Sign}(\mathbf{v}_t)\rangle - \eta\langle\nabla f(\mathbf{x}_t), \mathrm{Sign}(\nabla f(\mathbf{x}_t))\rangle + \frac{\eta^2 Ld}{2}$$
$$= f(\mathbf{x}_t) + \eta\langle\nabla f(\mathbf{x}_t), \mathrm{Sign}(\nabla f(\mathbf{x}_t)) - \mathrm{Sign}(\mathbf{v}_t)\rangle - \eta\|\nabla f(\mathbf{x}_t)\|_1 + \frac{\eta^2 Ld}{2}$$
$$= f(\mathbf{x}_t) + \eta\sum_{i=1}^{d}\langle[\nabla f(\mathbf{x}_t)]_i, \mathrm{Sign}([\nabla f(\mathbf{x}_t)]_i) - \mathrm{Sign}([\mathbf{v}_t]_i)\rangle - \eta\|\nabla f(\mathbf{x}_t)\|_1 + \frac{\eta^2 Ld}{2}$$
$$\leq f(\mathbf{x}_t) + \eta\sum_{i=1}^{d}2\,|[\nabla f(\mathbf{x}_t)]_i| \cdot \mathbb{I}\left(\mathrm{Sign}([\nabla f(\mathbf{x}_t)]_i) \neq \mathrm{Sign}([\mathbf{v}_t]_i)\right) - \eta\|\nabla f(\mathbf{x}_t)\|_1 \tag{13}$$
$$+ \frac{\eta^2 Ld}{2}$$
$$\leq f(\mathbf{x}_t) + \eta\sum_{i=1}^{d}2|[\nabla f(\mathbf{x}_t)]_i - [\mathbf{v}_t]_i| \cdot \mathbb{I}\left(\mathrm{Sign}([\nabla f(\mathbf{x}_t)]_i) \neq \mathrm{Sign}([\mathbf{v}_t]_i)\right) - \eta\|\nabla f(\mathbf{x}_t)\|_1$$
$$+ \frac{\eta^2 Ld}{2}$$
$$\leq f(\mathbf{x}_t) + \eta\sum_{i=1}^{d}2|[\nabla f(\mathbf{x}_t)]_i - [\mathbf{v}_t]_i| - \eta\|\nabla f(\mathbf{x}_t)\|_1 + \frac{\eta^2 Ld}{2}$$
$$= f(\mathbf{x}_t) + 2\eta\|\nabla f(\mathbf{x}_t) - \mathbf{v}_t\|_1 - \eta\|\nabla f(\mathbf{x}_t)\|_1 + \frac{\eta^2 Ld}{2}$$
$$\leq f(\mathbf{x}_t) + 2\eta\sqrt{d}\|\nabla f(\mathbf{x}_t) - \mathbf{v}_t\| - \eta\|\nabla f(\mathbf{x}_t)\|_1 + \frac{\eta^2 Ld}{2}.$$

Summing up and rearranging the equation (13), we derive:

$$\mathbb{E}\left[\frac{1}{T}\sum_{t=1}^{T}\|\nabla f(\mathbf{x}_t)\|_1\right]$$
$$\leq \frac{f(\mathbf{x}_1) - f(\mathbf{x}_{T+1})}{\eta T} + 2\sqrt{d} \cdot \mathbb{E}\left[\frac{1}{T}\sum_{t=1}^{T}\|\nabla f(\mathbf{x}_t) - \mathbf{v}_t\|\right] + \frac{\eta Ld}{2} \tag{14}$$
$$\leq \frac{\Delta_f}{\eta T} + 2\sqrt{d} \cdot \sqrt{\mathbb{E}\left[\frac{1}{T}\sum_{t=1}^{T}\|\nabla f(\mathbf{x}_t) - \mathbf{v}_t\|^2\right]} + \frac{\eta Ld}{2}.$$

where we define $\Delta_f = f(\mathbf{x}_1) - f_*$, and the second inequality is due to Jensen's Inequality.

Next, we can bound the term $\mathbb{E}\left[\frac{1}{T}\sum_{t=1}^{T}\|\nabla f(\mathbf{x}_t) - \mathbf{v}_t\|^2\right]$ as follows.

$$
\begin{aligned}
\mathbb{E}\left[\|\nabla f(\mathbf{x}_{t+1}) - \mathbf{v}_{t+1}\|^2\right] &= \mathbb{E}\left[\|(1-\beta)\mathbf{v}_t + \beta\nabla f(\mathbf{x}_{t+1};\xi_{t+1}) - \nabla f(\mathbf{x}_{t+1})\|^2\right] \\
&= \mathbb{E}\left[\|(1-\beta)(\mathbf{v}_t - \nabla f(\mathbf{x}_t)) + \beta\left(\nabla f(\mathbf{x}_{t+1};\xi_{t+1}) - \nabla f(\mathbf{x}_{t+1})\right)\right. \\
&\qquad \left. + (1-\beta)\left(\nabla f(\mathbf{x}_t) - \nabla f(\mathbf{x}_{t+1})\right)\|^2\right] \\
&= (1-\beta)^2\mathbb{E}\left[\|\mathbf{v}_t - \nabla f(\mathbf{x}_t) + \nabla f(\mathbf{x}_t) - \nabla f(\mathbf{x}_{t+1})\|^2\right] \\
&\qquad + \beta^2\mathbb{E}\left[\|\nabla f(\mathbf{x}_{t+1};\xi_{t+1}) - \nabla f(\mathbf{x}_{t+1})\|^2\right] \\
&\leq (1-\beta)^2(1+\beta)\mathbb{E}\left[\|\mathbf{v}_t - \nabla f(\mathbf{x}_t)\|^2\right] \\
&\qquad + (1-\beta)^2(1+\frac{1}{\beta})\mathbb{E}\left[\|\nabla f(\mathbf{x}_t) - \nabla f(\mathbf{x}_{t+1})\|^2\right] + \beta^2\sigma^2 \\
&\leq (1-\beta)\mathbb{E}\left[\|\mathbf{v}_t - \nabla f(\mathbf{x}_t)\|^2\right] + \frac{2L^2}{\beta}\|\mathbf{x}_{t+1} - \mathbf{x}_t\|^2 + \beta^2\sigma^2 \\
&\leq (1-\beta)\mathbb{E}\left[\|\mathbf{v}_t - \nabla f(\mathbf{x}_t)\|^2\right] + \frac{2\eta^2 L^2 d}{\beta} + \beta^2\sigma^2,
\end{aligned}
$$

where the third equality is due to the fact $\mathbb{E}\left[\nabla f(\mathbf{x}_{t+1};\xi_{t+1}) - \nabla f(\mathbf{x}_{t+1})\right] = 0$. Summing up, we can ensure

$$
\begin{aligned}
\mathbb{E}\left[\frac{1}{T}\sum_{t=1}^{T}\|\mathbf{v}_t - \nabla f(\mathbf{x}_t)\|^2\right] &\leq \frac{\mathbb{E}\left[\|\mathbf{v}_1 - \nabla f(\mathbf{x}_1)\|^2\right]}{\beta T} + \frac{2\eta^2 L^2 d}{\beta^2} + \beta\sigma^2 \\
&\leq \frac{\sigma^2}{\beta T} + \frac{2\eta^2 L^2 d}{\beta^2} + \beta\sigma^2.
\end{aligned}
\tag{15}
$$

Incorporating the above into equation (14) and setting that $\beta = \mathcal{O}\left(T^{-1/2}\right), \eta = \mathcal{O}\left(d^{-1/2}T^{-3/4}\right)$, we observe:

$$
\begin{aligned}
\mathbb{E}\left[\frac{1}{T}\sum_{t=1}^{T}\|\nabla f(\mathbf{x}_t)\|_1\right] &\leq \frac{\Delta_f}{\eta T} + 2\sqrt{d}\cdot\sqrt{\mathbb{E}\left[\frac{1}{T}\sum_{t=1}^{T}\|\nabla f(\mathbf{x}_t) - \mathbf{v}_t\|^2\right]} + \frac{\eta L d}{2} \\
&\leq \frac{\Delta_f}{\eta T} + 2\sqrt{d}\cdot\sqrt{\frac{\sigma^2}{\beta T} + \frac{2\eta^2 L^2 d}{\beta^2} + \beta\sigma^2} + \frac{\eta L d}{2} \\
&= \mathcal{O}\left(\frac{(1 + \Delta_f + \sigma + L)d^{1/2}}{T^{1/4}}\right) \\
&= \mathcal{O}\left(\frac{d^{1/2}}{T^{1/4}}\right),
\end{aligned}
$$

which finishes the proof of Theorem 2.

## C   PROOF OF THEOREM 3 AND 4

Since the overall objective function $f(\mathbf{x})$ is $L$-smooth, we have the following:

$$
\begin{aligned}
f(\mathbf{x}_{t+1}) \leq & f(\mathbf{x}_t) + \langle \nabla f(\mathbf{x}_t), \mathbf{x}_{t+1} - \mathbf{x}_t \rangle + \frac{L}{2}\|\mathbf{x}_{t+1} - \mathbf{x}_t\|^2 \\
\leq & f(\mathbf{x}_t) - \eta \left\langle \nabla f(\mathbf{x}_t), \mathrm{Sign}\left(\frac{1}{n}\sum_{j=1}^n \mathrm{S}_G(\mathbf{v}_t^j)\right)\right\rangle + \frac{\eta^2 L d}{2} \\
= & f(\mathbf{x}_t) + \eta \left\langle \nabla f(\mathbf{x}_t), \mathrm{Sign}(\nabla f(\mathbf{x}_t)) - \mathrm{Sign}\left(\frac{1}{n}\sum_{j=1}^n \mathrm{S}_G(\mathbf{v}_t^j)\right)\right\rangle \\
& - \eta \langle \nabla f(\mathbf{x}_t), \mathrm{Sign}(\nabla f(\mathbf{x}_t))\rangle + \frac{\eta^2 L d}{2} \\
= & f(\mathbf{x}_t) + \eta \left\langle \nabla f(\mathbf{x}_t), \mathrm{Sign}(\nabla f(\mathbf{x}_t)) - \mathrm{Sign}\left(\frac{1}{n}\sum_{j=1}^n \mathrm{S}_G(\mathbf{v}_t^j)\right)\right\rangle \\
& - \eta \|\nabla f(\mathbf{x}_t)\|_1 + \frac{\eta^2 L d}{2} \\
\leq & f(\mathbf{x}_t) + 2\eta R\sqrt{d} \left\|\frac{\nabla f(\mathbf{x}_t)}{R} - \frac{1}{n}\sum_{j=1}^n \mathrm{S}_G(\mathbf{v}_t^j)\right\| - \eta \|\nabla f(\mathbf{x}_t)\|_1 + \frac{\eta^2 L d}{2},
\end{aligned}
\tag{16}
$$

where the last inequality is because of

$$
\begin{aligned}
& \left\langle \nabla f(\mathbf{x}_t), \mathrm{Sign}(\nabla f(\mathbf{x}_t)) - \mathrm{Sign}\left(\frac{1}{n}\sum_{j=1}^n \mathrm{S}_G(\mathbf{v}_t^j)\right)\right\rangle \\
= & \sum_{i=1}^d \left\langle [\nabla f(\mathbf{x}_t)]_i, \mathrm{Sign}([\nabla f(\mathbf{x}_t)]_i) - \mathrm{Sign}\left(\left[\frac{1}{n}\sum_{j=1}^n \mathrm{S}_G(\mathbf{v}_t^j)\right]_i\right)\right\rangle \\
\leq & \sum_{i=1}^d 2 \left|[\nabla f(\mathbf{x}_t)]_i\right| \cdot \mathbb{I}\left(\mathrm{Sign}([\nabla f(\mathbf{x}_t)]_i) \neq \mathrm{Sign}\left(\left[\frac{1}{n}\sum_{j=1}^n S(\mathbf{v}_t^j)\right]_i\right)\right) \\
\leq & \sum_{i=1}^d 2R \left|\frac{[\nabla f(\mathbf{x}_t)]_i}{R}\right| \cdot \mathbb{I}\left(\mathrm{Sign}([\nabla f(\mathbf{x}_t)]_i) \neq \mathrm{Sign}\left(\left[\frac{1}{n}\sum_{j=1}^n \mathrm{S}_G(\mathbf{v}_t^j)\right]_i\right)\right) \\
\leq & \sum_{i=1}^d 2R \left|\frac{[\nabla f(\mathbf{x}_t)]_i}{R} - \left[\frac{1}{n}\sum_{j=1}^n \mathrm{S}_G(\mathbf{v}_t^j)\right]_i\right| \cdot \mathbb{I}\left(\mathrm{Sign}([\nabla f(\mathbf{x}_t)]_i) \neq \mathrm{Sign}\left(\left[\frac{1}{n}\sum_{j=1}^n \mathrm{S}_G(\mathbf{v}_t^j)\right]_i\right)\right) \\
\leq & \sum_{i=1}^d 2R \left|\frac{[\nabla f(\mathbf{x}_t)]_i}{R} - \left[\frac{1}{n}\sum_{j=1}^n \mathrm{S}_G(\mathbf{v}_t^j)\right]_i\right| \\
= & 2R \left\|\frac{\nabla f(\mathbf{x}_t)}{R} - \frac{1}{n}\sum_{j=1}^n \mathrm{S}_G(\mathbf{v}_t^j)\right\|_1 \\
\leq & 2R\sqrt{d} \left\|\frac{\nabla f(\mathbf{x}_t)}{R} - \frac{1}{n}\sum_{j=1}^n \mathrm{S}_G(\mathbf{v}_t^j)\right\|.
\end{aligned}
\tag{17}
$$

Rearranging and taking the expectation over equation (16), we have:

$$\mathbb{E}\left[f(\mathbf{x}_{t+1}) - f(\mathbf{x}_t)\right]$$

$$\leq 2\eta G\sqrt{d}\mathbb{E}\left[\left\|\frac{\nabla f(\mathbf{x}_t)}{G} - \frac{1}{n}\sum_{j=1}^{n}\mathrm{S}_G(\mathbf{v}_t^j)\right\|\right] - \eta\mathbb{E}\left[\|\nabla f(\mathbf{x}_t)\|_1\right] + \frac{\eta^2 Ld}{2}$$

$$\leq 2\eta G\sqrt{d}\mathbb{E}\left[\left\|\frac{\nabla f(\mathbf{x}_t)}{G} - \frac{1}{nG}\sum_{j=1}^{n}\mathbf{v}_t^j\right\|\right] + 2\eta G\sqrt{d}\mathbb{E}\left[\left\|\frac{1}{n}\sum_{j=1}^{n}\left(\mathrm{S}_G(\mathbf{v}_t^j) - \frac{\mathbf{v}_t^j}{G}\right)\right\|\right]$$

$$- \eta\mathbb{E}\left[\|\nabla f(\mathbf{x}_t)\|_1\right] + \frac{\eta^2 Ld}{2}$$

$$\leq 2\eta\sqrt{d}\mathbb{E}\left[\left\|\nabla f(\mathbf{x}_t) - \frac{1}{n}\sum_{j=1}^{n}\mathbf{v}_t^j\right\|\right] + 2\eta G\sqrt{d}\sqrt{\mathbb{E}\left[\left\|\frac{1}{n}\sum_{j=1}^{n}\left(\mathrm{S}_G(\mathbf{v}_t^j) - \frac{\mathbf{v}_t^j}{G}\right)\right\|^2\right]}$$

$$- \eta\mathbb{E}\left[\|\nabla f(\mathbf{x}_t)\|_1\right] + \frac{\eta^2 Ld}{2} \tag{18}$$

$$\leq 2\eta\sqrt{d}\mathbb{E}\left[\left\|\nabla f(\mathbf{x}_t) - \frac{1}{n}\sum_{j=1}^{n}\mathbf{v}_t^j\right\|\right] + 2\eta G\sqrt{d}\sqrt{\frac{1}{n^2}\sum_{j=1}^{n}\mathbb{E}\left[\left\|\left(\mathrm{S}_G(\mathbf{v}_t^j) - \frac{\mathbf{v}_t^j}{G}\right)\right\|^2\right]}$$

$$- \eta\mathbb{E}\left[\|\nabla f(\mathbf{x}_t)\|_1\right] + \frac{\eta^2 Ld}{2}$$

$$\leq 2\eta\sqrt{d}\mathbb{E}\left[\left\|\nabla f(\mathbf{x}_t) - \frac{1}{n}\sum_{j=1}^{n}\mathbf{v}_t^j\right\|\right] + 2\eta G\sqrt{d}\sqrt{\frac{1}{n^2}\sum_{j=1}^{n}\mathbb{E}\left[\left\|\mathrm{S}_G(\mathbf{v}_t^j)\right\|^2\right]}$$

$$- \eta\mathbb{E}\left[\|\nabla f(\mathbf{x}_t)\|_1\right] + \frac{\eta^2 Ld}{2}$$

$$\leq 2\eta\sqrt{d}\mathbb{E}\left[\left\|\nabla f(\mathbf{x}_t) - \frac{1}{n}\sum_{j=1}^{n}\mathbf{v}_t^j\right\|\right] + \frac{2\eta dG}{\sqrt{n}} - \eta\mathbb{E}\left[\|\nabla f(\mathbf{x}_t)\|_1\right] + \frac{\eta^2 Ld}{2},$$

where the third inequality is due to the fact that $(\mathbb{E}\left[X\right])^2 \leq \mathbb{E}\left[X^2\right]$, and the forth inequality is because of $\mathbb{E}\left[S_G\left(\mathbf{v}_t^j\right)\right] = \frac{\mathbf{v}_t^j}{G}$, as well as the $S_G$ operation in each node is independent.

Rearranging the terms and summing up, we have:

$$\frac{1}{T}\sum_{i=1}^{T}\mathbb{E}\left[\|\nabla f(\mathbf{x}_t)\|_1\right] \leq \frac{\Delta_f}{\eta T} + 2\sqrt{d}\mathbb{E}\left[\frac{1}{T}\sum_{i=1}^{T}\left\|\nabla f(\mathbf{x}_t) - \frac{1}{n}\sum_{j=1}^{n}\mathbf{v}_t^j\right\|\right] + \frac{2dG}{\sqrt{n}} + \frac{\eta Ld}{2}$$

$$\leq \frac{\Delta_f}{\eta T} + 2\sqrt{d}\sqrt{\mathbb{E}\left[\frac{1}{T}\sum_{i=1}^{T}\left\|\nabla f(\mathbf{x}_t) - \frac{1}{n}\sum_{j=1}^{n}\mathbf{v}_t^j\right\|^2\right]} + \frac{2dG}{\sqrt{n}} + \frac{\eta Ld}{2},$$

where the last inequality is due to Jensen's inequality.

For each worker $j$, we have the following according to the definition of $\mathbf{v}_t^j$:

$$\mathbf{v}_{t+1}^j - \nabla f_j(\mathbf{x}_{t+1}) = (1-\beta)\left(\mathbf{v}_t^j - \nabla f_j(\mathbf{x}_t)\right) + \beta\left(\nabla f_j(\mathbf{x}_{t+1}; \xi_{t+1}^j) - \nabla f_j(\mathbf{x}_{t+1})\right)$$

$$+ (1-\beta)\left(\nabla f_j(\mathbf{x}_t) - \nabla f_j(\mathbf{x}_{t+1})\right).$$

Averaging over $\{n\}$ and noting that $\nabla f(\mathbf{x}) = \frac{1}{n}\sum_{j=1}^n \nabla f_j(\mathbf{x})$, we can obtain:

$$\frac{1}{n}\sum_{j=1}^n \mathbf{v}_{t+1}^j - \nabla f(\mathbf{x}_{t+1}) = \frac{1}{n}\sum_{j=1}^n \left(\mathbf{v}_{t+1}^j - \nabla f_j(\mathbf{x}_{t+1})\right)$$

$$= (1-\beta)\frac{1}{n}\sum_{j=1}^n \left(\mathbf{v}_t^j - \nabla f_j(\mathbf{x}_t)\right) + \beta \frac{1}{n}\sum_{j=1}^n \left(\nabla f_j(\mathbf{x}_{t+1};\xi_{t+1}^j) - \nabla f_j(\mathbf{x}_{t+1})\right)$$

$$+ (1-\beta)\frac{1}{n}\sum_{j=1}^n \left(\nabla f_j(\mathbf{x}_t) - \nabla f_j(\mathbf{x}_{t+1})\right).$$

Then we have

$$\mathbb{E}\left[\left\|\frac{1}{n}\sum_{j=1}^n \mathbf{v}_{t+1}^j - \nabla f(\mathbf{x}_{t+1})\right\|^2\right]$$

$$\leq (1-\beta)\mathbb{E}\left[\left\|\frac{1}{n}\sum_{j=1}^n \left(\mathbf{v}_t^j - \nabla f_j(\mathbf{x}_t)\right)\right\|^2\right] + \beta^2 \frac{1}{n^2}\sum_{j=1}^n \mathbb{E}\left[\left\|\nabla f_j(\mathbf{x}_{t+1};\xi_{t+1}^j) - \nabla f_j(\mathbf{x}_{t+1})\right\|^2\right]$$

$$+ \frac{2}{\beta n}\sum_{j=1}^n \mathbb{E}\left[\left\|\nabla f_j(\mathbf{x}_{t+1}) - \nabla f_j(\mathbf{x}_t)\right\|^2\right]$$

$$\leq (1-\beta)\mathbb{E}\left[\left\|\frac{1}{n}\sum_{j=1}^n \left(\mathbf{v}_t^j - \nabla f_j(\mathbf{x}_t)\right)\right\|^2\right] + \frac{\beta^2\sigma^2}{n} + \frac{2L^2}{\beta}\left\|\mathbf{x}_{t+1} - \mathbf{x}_t\right\|^2$$

$$\leq (1-\beta)\mathbb{E}\left[\left\|\frac{1}{n}\sum_{j=1}^n \mathbf{v}_t^j - \nabla f(\mathbf{x}_t)\right\|^2\right] + \frac{\beta^2\sigma^2}{n} + \frac{2L^2\eta^2 d}{\beta}.$$

By summing up and rearranging, we observe

$$\mathbb{E}\left[\frac{1}{T}\sum_{t=1}^T \left\|\frac{1}{n}\sum_{j=1}^n \mathbf{v}_t^j - \nabla f(\mathbf{x}_t)\right\|^2\right] \leq \frac{\mathbb{E}\left[\left\|\frac{1}{n}\sum_{j=1}^n \mathbf{v}_1^j - \nabla f(\mathbf{x}_1)\right\|^2\right]}{\beta T} + \frac{\beta\sigma^2}{n} + \frac{2L^2\eta^2 d}{\beta^2} \quad (19)$$

$$\leq \frac{\sigma^2}{n\beta T} + \frac{\sigma^2\beta}{n} + \frac{2L^2\eta^2 d}{\beta^2}.$$

Finally, we can ensure that

$$\frac{1}{T}\sum_{i=1}^T \|\nabla f(\mathbf{x}_t)\|_1 \leq \frac{\Delta_f}{\eta T} + \frac{2dG}{\sqrt{n}} + \frac{\eta Ld}{2} + 2\sqrt{d}\sqrt{\mathbb{E}\left[\frac{1}{T}\sum_{i=1}^T \left\|\nabla f(\mathbf{x}_t) - \frac{1}{n}\sum_{j=1}^n \mathbf{v}_t^j\right\|^2\right]}$$

$$\leq \frac{\Delta_f}{\eta T} + \frac{2dG}{\sqrt{n}} + \frac{\eta Ld}{2} + 2\sqrt{d}\sqrt{\frac{\sigma^2}{n\beta T} + \frac{\sigma^2\beta}{n} + \frac{2L^2\eta^2 d}{\beta^2}}.$$

By setting $\beta = \frac{1}{2}$ and $\eta = \mathcal{O}\left(T^{-1/2}d^{-1/2}\right)$, we have

$$\frac{1}{T}\sum_{i=1}^T \|\nabla f(\mathbf{x}_t)\|_1 = \mathcal{O}\left(\frac{d^{1/2}}{T^{1/2}} + \frac{d}{n^{1/2}}\right).$$

By setting $\beta = \frac{1}{2}$ and $\eta = \mathcal{O}\left(n^{-1/2}\right)$, we have

$$\frac{1}{T}\sum_{i=1}^T \|\nabla f(\mathbf{x}_t)\|_1 = \mathcal{O}\left(\frac{n^{1/2}}{T} + \frac{d}{n^{1/2}}\right).$$

## D   PROOF OF THEOREM 5

Due to the fact that the overall objective function $f(\mathbf{x})$ is $L$-smooth, we have the following:

$$
\begin{aligned}
f(\mathbf{x}_{t+1}) \leq & f(\mathbf{x}_t) + \langle \nabla f(\mathbf{x}_t), \mathbf{x}_{t+1} - \mathbf{x}_t \rangle + \frac{L}{2} \|\mathbf{x}_{t+1} - \mathbf{x}_t\|^2 \\
\leq & f(\mathbf{x}_t) - \eta \left\langle \nabla f(\mathbf{x}_t), \mathrm{S}_1 \left( \frac{1}{n} \sum_{j=1}^n \mathrm{S}_G(\mathbf{v}_t^j) \right) \right\rangle + \frac{\eta^2 L d}{2} \\
= & f(\mathbf{x}_t) + \eta \left\langle \nabla f(\mathbf{x}_t), \frac{\nabla f(\mathbf{x}_t)}{G} - \mathrm{S}_1 \left( \frac{1}{n} \sum_{j=1}^n \mathrm{S}_G(\mathbf{v}_t^j) \right) \right\rangle \\
& - \eta \left\langle \nabla f(\mathbf{x}_t), \frac{\nabla f(\mathbf{x}_t)}{G} \right\rangle + \frac{\eta^2 L d}{2} \\
= & f(\mathbf{x}_t) + \eta \left\langle \nabla f(\mathbf{x}_t), \frac{\nabla f(\mathbf{x}_t)}{G} - \mathrm{S}_1 \left( \frac{1}{n} \sum_{j=1}^n \mathrm{S}_G(\mathbf{v}_t^j) \right) \right\rangle - \frac{\eta}{G} \|\nabla f(\mathbf{x}_t)\|^2 + \frac{\eta^2 L d}{2}.
\end{aligned}
$$

Taking expectations leads to:

$$
\begin{aligned}
& \mathbb{E}\left[ f(\mathbf{x}_{t+1}) - f(\mathbf{x}_t) \right] \\
\leq & \eta \mathbb{E}\left[ \left\langle \nabla f(\mathbf{x}_t), \frac{1}{G} \nabla f(\mathbf{x}_t) - \mathrm{S}_1 \left( \frac{1}{n} \sum_{j=1}^n \mathrm{S}_G(\mathbf{v}_t^j) \right) \right\rangle \right] - \frac{\eta}{G} \mathbb{E}\left[ \|\nabla f(\mathbf{x}_t)\|^2 \right] + \frac{\eta^2 L d}{2} \\
= & \eta \mathbb{E}\left[ \left\langle \nabla f(\mathbf{x}_t), \frac{1}{G} \nabla f(\mathbf{x}_t) - \frac{1}{n} \sum_{j=1}^n \mathrm{S}_G(\mathbf{v}_t^j) \right\rangle \right] - \frac{\eta}{G} \mathbb{E}\left[ \|\nabla f(\mathbf{x}_t)\|^2 \right] + \frac{\eta^2 L d}{2} \\
= & \eta \mathbb{E}\left[ \left\langle \nabla f(\mathbf{x}_t), \frac{1}{G} \nabla f(\mathbf{x}_t) - \frac{1}{nG} \sum_{j=1}^n \mathbf{v}_t^j \right\rangle \right] - \frac{\eta}{G} \mathbb{E}\left[ \|\nabla f(\mathbf{x}_t)\|^2 \right] + \frac{\eta^2 L d}{2} \quad (20) \\
\leq & \eta \mathbb{E}\left[ \frac{1}{2G} \|\nabla f(\mathbf{x}_t)\|^2 + \frac{1}{2G} \left\| \nabla f(\mathbf{x}_t) - \frac{1}{n} \sum_{j=1}^n \mathbf{v}_t^j \right\|^2 \right] - \frac{\eta}{G} \mathbb{E}\left[ \|\nabla f(\mathbf{x}_t)\|^2 \right] + \frac{\eta^2 L d}{2} \\
= & \frac{\eta}{2G} \mathbb{E}\left[ \left\| \nabla f(\mathbf{x}_t) - \frac{1}{n} \sum_{j=1}^n \mathbf{v}_t^j \right\|^2 \right] - \frac{\eta}{2G} \mathbb{E}\left[ \|\nabla f(\mathbf{x}_t)\|^2 \right] + \frac{\eta^2 L d}{2}.
\end{aligned}
$$

Rearranging the terms and summing up:

$$
\begin{aligned}
\frac{1}{T} \sum_{i=1}^T \mathbb{E}\left\| \nabla f(\mathbf{x}_t) \right\|^2 \leq & \frac{2\Delta_f G}{\eta T} + \mathbb{E}\left[ \frac{1}{T} \sum_{i=1}^T \left\| \nabla f(\mathbf{x}_t) - \frac{1}{n} \sum_{j=1}^n \mathbf{v}_t^j \right\|^2 \right] + \eta L d G \\
\leq & \frac{2\Delta_f G}{\eta T} + \mathbb{E}\left[ \frac{1}{n} \sum_{j=1}^n \frac{1}{T} \sum_{i=1}^T \left\| \nabla f_j(\mathbf{x}_t) - \mathbf{v}_t^j \right\|^2 \right] + \eta L d G.
\end{aligned}
$$

For each worker $j$, according to the definition of $\mathbf{v}_t^j$, we have:

$$
\begin{aligned}
\mathbf{v}_{t+1}^j - \nabla f_j(\mathbf{x}_{t+1}) = & (1 - \beta) \left( \mathbf{v}_t^j - \nabla f_j(\mathbf{x}_t) \right) + \beta \left( \nabla f_j(\mathbf{x}_{t+1}; \xi_{t+1}^j) - \nabla f_j(\mathbf{x}_{t+1}) \right) \\
& + (1 - \beta) \left( \nabla f_j(\mathbf{x}_t) - \nabla f_j(\mathbf{x}_{t+1}) \right).
\end{aligned}
$$

Table 5: Optimal hyperparameters for fine-tuning GPT-2 on Alpaca.

| Method | SGDM | signSGD | EF-signSGD | AdamW | Signum |
|---|---|---|---|---|---|
| lr | 1e-1 | 1e-4 | 1e-0 | 5e-4 | 1e-4 |
| $\beta_1$ | 0.9 | – | – | 0.9 | 0.75 |
| $\beta_2$ | – | – | – | 0.95 | – |

Table 6: Optimal hyperparameters for fine-tuning Qwen3-0.6B on Alpaca.

| Method | SGDM | signSGD | EF-signSGD | AdamW | Signum |
|---|---|---|---|---|---|
| lr | 5e-2 | 1e-5 | 5e-1 | 5e-5 | 1e-5 |
| $\beta_1$ | 0.9 | – | – | 0.9 | 0.75 |
| $\beta_2$ | – | – | – | 0.95 | – |

Then we have

$$\mathbb{E}\left[\left\|\mathbf{v}_{t+1}^j - \nabla f_j(\mathbf{x}_{t+1})\right\|^2\right]$$

$$\leq (1-\beta)\mathbb{E}\left[\left\|\mathbf{v}_t^j - \nabla f_j(\mathbf{x}_t)\right\|^2\right] + \beta^2\mathbb{E}\left[\left\|\nabla f_j(\mathbf{x}_{t+1}; \xi_{t+1}^j) - \nabla f_j(\mathbf{x}_{t+1})\right\|^2\right]$$

$$+ \frac{2}{\beta}\mathbb{E}\left[\|\nabla f_j(\mathbf{x}_{t+1}) - \nabla f_j(\mathbf{x}_t)\|^2\right]$$

$$\leq (1-\beta)\mathbb{E}\left[\left\|\mathbf{v}_t^j - \nabla f_j(\mathbf{x}_t)\right\|^2\right] + \beta^2\sigma^2 + \frac{2L^2}{\beta}\|\mathbf{x}_{t+1} - \mathbf{x}_t\|^2$$

$$\leq (1-\beta)\mathbb{E}\left[\left\|\mathbf{v}_t^j - \nabla f_j(\mathbf{x}_t)\right\|^2\right] + \beta^2\sigma^2 + \frac{2L^2\eta^2 d}{\beta}.$$

As a result, we know that

$$\mathbb{E}\left[\frac{1}{n}\sum_{j=1}^n \frac{1}{T}\sum_{t=1}^T \left\|\mathbf{v}_t^j - \nabla f_j(\mathbf{x}_t)\right\|^2\right] \leq \frac{\sigma^2}{\beta T} + \sigma^2\beta + \frac{2L^2\eta^2 d}{\beta^2}.$$

Finally, we can obtain the final bound:

$$\mathbb{E}\left[\frac{1}{T}\sum_{i=1}^T \|\nabla f(\mathbf{x}_t)\|\right] \leq \sqrt{\mathbb{E}\left[\frac{1}{T}\sum_{i=1}^T \|\nabla f(\mathbf{x}_t)\|^2\right]}$$

$$\leq \sqrt{\frac{2\Delta_f G}{\eta T} + \eta L d G + \frac{\sigma^2}{\beta T} + \sigma^2\beta + \frac{2L^2\eta^2 d}{\beta^2}}.$$

That is to say, by setting $\beta = \eta^{2/3}d^{1/3}$, $\eta = \mathcal{O}\left(\min\left\{\frac{1}{T^{1/2}d^{1/2}}, \frac{1}{T^{3/5}d^{1/5}}\right\}\right)$, we can obtain the convergence rate of $\mathcal{O}\left(\max\left\{\frac{d^{1/4}}{T^{1/4}}, \frac{d^{1/10}}{T^{1/5}}\right\}\right)$.

## E EXPERIMENTAL DETAILS

In this section, we present the omitted details in our experiments.

### E.1 OPTIMAL HYPERPARAMETERS

The tuned learning rates and $\beta_1, \beta_2$ coefficients for all methods are shown in Tables 5 and 6, which can be used to reproduce the results in Table 4. We underline that our tuned optimal learning rate of

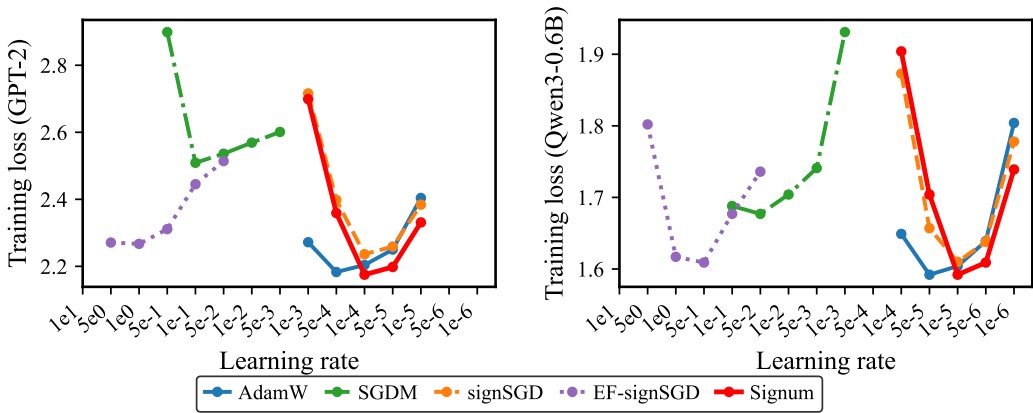

Figure 3: Sensitivity result across different learning rates.

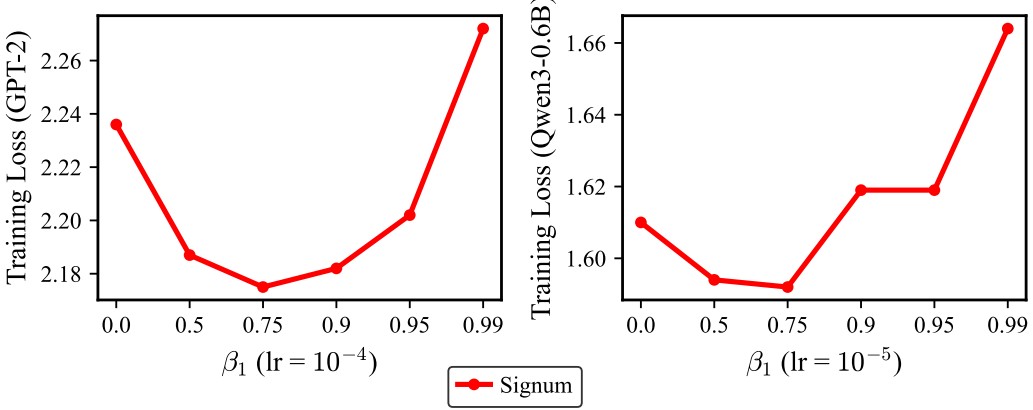

Figure 4: Sensitivity result across different momentum coefficients.

AdamW is **5 times** that of Signum, which aligns perfectly well with the theoretical value (Kosson et al., 2024), as well as the recent empirical discovery (Liu et al., 2025a)[1].

### E.2 SENSITIVITY ANALYSIS

Then, we conduct a sensitivity analysis across different learning rates and different momentum coefficients $\beta_1$ (representing $1 - \beta$ in Algorithm 1). We keep the other hyperparameters fixed to their optimal values and vary only the learning rate or the momentum coefficient to see the changes in training loss values. Figure 3 shows the training losses on GPT-2 and Qwen3-0.6B for all methods across a wide range of learning rates. Our method remains valid and stable within a certain range. While we observe that AdamW is less sensitive to learning rate changes, this is largely due to its smaller update RMS norm (Liu et al., 2025a). We also investigate the sensitivity of $\beta_1$ in Signum, as shown in Figure 4. The results demonstrate that Signum remains quite robust to different momentum coefficients.

---

[1]Such phenomenon stems from the idea of matching update RMS norms between sign-based methods and AdamW (Liu et al., 2025a). Signum has an inherent update RMS norm of 1, while the value of AdamW typically ranges from 0.2 to 0.4 (Liu et al., 2025a;b) with theoretical estimation of $\sqrt{(1 - \beta_1)/(1 + \beta_1)}$ (Kosson et al., 2024). Matching these terms (1 VS 0.2) gives a rough law of $\texttt{lr}_{\text{AdamW}} \approx 5\texttt{lr}_{\text{Signum}}$.

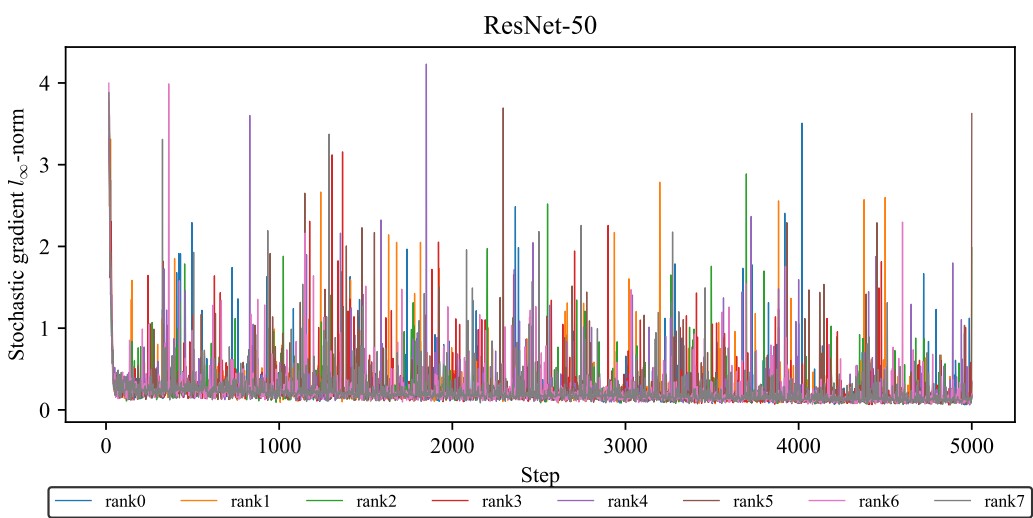

Figure 5: Stochastic gradient trajectory on the CIFAR-100 dataset.

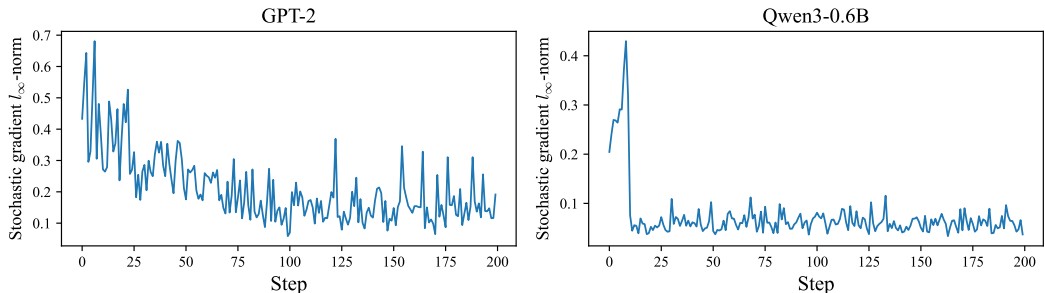

Figure 6: Stochastic gradient trajectory on the Alpaca dataset.

### E.3 EMPIRICAL VALIDATION OF BOUNDED STOCHASTIC GRADIENTS

We investigate the infinity norm of stochastic gradients along the training trajectory. The direct evidence in Figure 5 shows that Assumption 8 is well-satisfied in distributed environments. We also consider the centralized environment (where the number of nodes $n = 1$), whose trend is depicted in Figure 6. It is evident that the stochastic gradients are bounded in both scenarios, highlighting the rationality of Assumption 8.

