# OpenReview forum: "Improved Analysis for Sign-based Methods with Momentum Updates"
_ICLR.cc/2026/Conference — Submitted to ICLR 2026_

### Official Review · Reviewer_fqxZ · 2025-10-15

**Soundness:** 2
**Presentation:** 2
**Contribution:** 2
**Rating:** 4
**Confidence:** 3

**Summary:**

The paper presents a new analysis for sign-momentum. First, it presents a convergence rate of $O(\frac{\sqrt{\|\mathbf{L}\|_1+\Delta}+\|\mathbf{\sigma}\|_1}{T^{1/4}})$ for the centralized case under a separable smoothness assumption. Next, for the standard smoothness assumption, it presents a convergence rate of  $O(\frac{d^{1/2}(L+\Delta+\sigma)}{T^{1/4}})$. In the distributed setup, it achieves better results than (Sun et al., 2023; Jin et al., 2021). The paper provides experimental results on CIFAR-10 for the proposed approaches against existing baselines.

**Strengths:**

* The theoretical results appear to improve upon existing work, although I am less familiar with this literature.
* The results on CIFAR-10 look good.

**Weaknesses:**

* The $S_G$ procedure depends on unknown parameters. It is also unclear how the authors implemented $S_G$ and which version was used in the distributed setup. Could the authors provide more details? It would also be helpful if they compared results across the proposed versions (v1 and v2) to support the theoretical claims better.
* Regarding Theorem 5, the idea of applying an unbiased estimation twice is interesting. However, in this case, shouldn’t we also observe a reduction in stochastic variance with the number of workers and, consequently, accelerated convergence, as is typically seen in standard distributed learning?
* The authors did not provide any supplementary material. Since no code for the empirical evaluation is available, it is difficult for the community to verify and assess the contribution of the proposed approach.

**Questions:**

See above.

---

> ### Author Response · Authors · 2025-11-20
>
> Thank you very much for your constructive comments and suggestions! We have addressed the concerns you raised via the following response. We sincerely hope that the reviewer can examine them and reevaluate our paper.
>
> >W1: The $S_G$ procedure depends on unknown parameters. It is also unclear how the authors implemented $S_G$ and which version was used in the distributed setup. Could the authors provide more details? It would also be helpful if they compared results across the proposed versions (v1 and v2) to support the theoretical claims better.
>
> **Response**: We apologize for the lack of clarity. Considering that we require $G \ge \Vert v \Vert_\infty$ to ensure the operation makes sense, we simply set $G = \Vert v_t \Vert_\infty$ for the current input vector $v_t$.
>
> MVSM (v1) vs (v2) in Experiments:
> The MVSM results shown in Figure 2 are actually MVSM-v2, and the results of MVSM-v1 are reported under the name `MV-signSGD-SIM' (we emphasized in the paper that MVSM-v1 is identical to MV-signSGD-SIM (with $\alpha = 0$)). It can be seen that MVSM-v2 is better than MVSM-v1 (i.e., MV-signSGD-SIM) in most cases. We have already clarified this in our revised paper.
>
> Comparison of MVSM (v1) and (v2): MVSM-v1 uses the traditional biased sign operation on the server, while MVSM-v2 employs the unbiased sign operation. As for theoretical guarantees, MVSM-v1 converges to $\mathcal{O}\left(\frac{d}{n^{1/2}}\right)$ when $T \to \infty$, but MVSM-v2 can converge to $0$ with $T \to \infty$. In the experiments in Figure 2, we also find that the gradient norm of MVSM-v1 is much larger than that of MVSM-v2 in the final steps, supporting our theoretical claims.
>
> >W2: Regarding Theorem 5, the idea of applying an unbiased estimation twice is interesting. However, in this case, shouldn’t we also observe a reduction in stochastic variance with the number of workers and, consequently, accelerated convergence, as is typically seen in standard distributed learning?
>
> **Response**: It is an interesting question to investigate. First, we have to emphasize that the existing sign-based methods can not ensure linear speedup. Note that even for the simple homogeneous setting, where the data distribution and the loss function on each node are the same, existing methods (Bernstein et al., 2018; 2019) can only obtain the convergence rate of $\mathcal{O}\left(\frac{1}{T^{1/4}}\left(1+\frac{1}{n} \right) \right)$, where $n$ is the number of nodes. That is to say, even in the homogeneous setting where $n$ workers contribute to learn the same distribution, previous literature still fails to obtain the linear speedup convergence rate of $\mathcal{O}\left(\frac{1}{(nT)^{1/4}}\right)$.
>
> When it comes to the heterogeneous environments investigated in this paper, the problem becomes more challenging since the data distribution in each worker can be very different, and thus, the learning problem is more complicated.
> Although more workers (larger $n$) could provide more information, this increase is largely weakened by only allowing transmitting the sign information in two directions, e.g., $x_{t+1} = x_{t} - \eta \operatorname{Sign}(\frac{1}{n} \sum_{j=1}^n \operatorname{Sign}(\cdot) )$. Also note that the outer sign operation itself introduces the variance, which can not be alleviated by using a larger $n$. We think this may be the main reason why we cannot ensure linear speedup for sign-based methods.
>
> >W3: The authors did not provide any supplementary material. Since no code for the empirical evaluation is available, it is difficult for the community to verify and assess the contribution of the proposed approach.
>
> **Response**: According to your request, we have made our code available via the following anonymous link: https://anonymous.4open.science/r/ICLR2026_Submission15147-DCFE.

---

> > ### Comment · Reviewer_fqxZ · 2025-11-27
> >
> > Thank you for the rebuttal. The authors have addressed my concerns, and I will raise my score.

---

> > > ### Author Response · Authors · 2025-11-28
> > >
> > > Thank you very much for your kind reply! We will revise our paper according to the constructive reviews.
> > >
> > > Best regards,
> > >
> > > Authors

---

### Official Review · Reviewer_2GAT · 2025-10-31

**Soundness:** 3
**Presentation:** 3
**Contribution:** 3
**Rating:** 6
**Confidence:** 2

**Summary:**

The paper shows that signSGD with momentum can achieve a convergence rate of $\mathcal{O}(T^{-1/4})$ using constant batch sizes without additional assumptions such as large batch sizes or unimodal symmetric stochastic noise. Experimental results are shown on CIFAR to demonstrate convergence rates.

**Strengths:**

Original tighter error analysis:
Bounds $\sum_i |[\nabla f(x_t)]_i| \cdot \mathbb{P}(\text{sign mismatch})$ directly by $\|\nabla f(x_t) - v_t\|_1$ instead of probability inequalities requiring $\mathcal{O}(\sqrt{T})$ batches or symmetric noise assumptions

Removes assumptions:
No $\mathcal{O}(\sqrt{T})$ batches or symmetric noise for $\mathcal{O}(T^{-1/4})$ rate that are required for prior analysis

Experimental validation:
Experiments with ResNet on CIFAR dataset show faster gradient norm decay that matches theoretical results

**Weaknesses:**

Incremental improvements:
The improvements are incremental rather than paradigm shifting. The improved bound is useful but may not have a large practical impact

Weak experimental results:
The experimental results are with ResNet on CIFAR datasets. These do not reflect modern uses cases of Sign-based methods. Experimental results on large models would make the paper stronger.

**Questions:**

How would the method perform on a larger model and more realistic datasets?

Assumption 8 $\sup_x \|\nabla f_j(x;\xi)\|_\infty \leq G$ is strong for neural networks, even though the authors cite previous works that use similar assumptions. Do your experiments satisfy this? What are observed \max_{t,j} \|\nabla f_j(x_t;\xi)\|_\infty values?

---

> ### Author Response · Authors · 2025-11-20
>
> Thank you very much for your constructive comments and suggestions!
>
> >W1: Incremental improvements: The improvements are incremental rather than paradigm shifting. The improved bound is useful but may not have a large practical impact.
>
> **Response**: We have to emphasize that previous analyses of signSGD or Signum either (i) require huge batch sizes on the order of $\mathcal{O}(\sqrt{T})$ (Bernstein et al., 2018), or (ii) rely on restrictive unimodal and symmetric noise assumptions (Bernstein et al., 2019), which are not realistic for common neural network gradients.
> In contrast, our work achieves the $\mathcal{O}\left(T^{-1/4}\right)$ rate without these additional assumptions. Furthermore, under $l_2$-smoothness, we accelerate the convergence rate from $\mathcal{O}\left( d T^{-1/4}\right)$ to $\mathcal{O}\left( d^{1/2} T^{-1/4}\right)$, which improve the rate by a factor of $\mathcal{O}\left(d^{1/2}\right)$.
> Considering that $d$ can be large for modern models, this improvement is highly meaningful in practice. Thus, we believe our contribution is significant, as we show that Signum works well without relying on the unrealistic assumptions that have traditionally been used to analyze it. Moreover, we introduce new algorithms for distributed settings (e.g., MVSM-v2) and achieve significant improvements in convergence rates (Theorems 3,4,5) compared to previous methods (Sun et al., 2023; Jin et al., 2021).
>
> To show the practical improvement, we have conducted additional experiments on fine-tuning the GPT-2 and Qwen3 models (see Section 5.3 of our revised paper), further highlighting the superiority of our method.
>
> >W2 \& Q1: Weak experimental results: The experimental results are with ResNet on CIFAR datasets. These do not reflect modern uses cases of Sign-based methods. Experimental results on large models would make the paper stronger. How would the method perform on a larger model and more realistic datasets?
>
> **Response**: We agree that experiments on larger-scale models would strengthen the work, and we note that sign-based methods have been employed for training large language models in recent research. In response to your suggestion, we have added new experiments on larger models, specifically training the GPT-2 and Qwen3 models on the Alpaca dataset. The results show that our Signum algorithm continues to outperform other methods (i.e., signSGD, SGDM, AdamW, and EF-SignSGD) when applied to training language models.
>
> >Q2: Assumption 8 $\sup_x |\nabla f_j(x;\xi)|\infty \leq G$ is strong for neural networks, even though the authors cite previous works that use similar assumptions. Do your experiments satisfy this? What are observed $\max_{t,j} |\nabla f_j(x_t;\xi)|_\infty$ values?
>
> **Response**: We first emphasize that Assumption 8 is required only for the distributed analysis, not for the centralized results. Then, we have to note that the bounded gradient assumption is standard and widely used in the theoretical study of sign-based methods in the heterogeneous setting (e.g., Jin et al., 2021; Sun et al., 2023). Additionally, our Assumption 8 is strictly weaker than that used in the prior work (Sun et al., 2023) we improve upon, which requires a bounded $l_2$-norm gradient.
>
> Furthermore, for many smooth loss functions (e.g., quadratic loss), gradients can be bounded as long as the domain is bounded. Since we update the variable $x_t$ using sign information, i.e., $x_t = x_{t-1} - \eta \cdot \text{sign}(\cdot)$, the update quantity would not be too large with a small learning rate. Therefore, the parameter vector would remain within a bounded region, and many functions are Lipschitz continuous within this domain.
>
> In practice, neural network gradients are typically bounded with modern training strategies such as standard initialization, data normalization, batch normalization, and gradient clipping. To validate this, we monitored the empirical maxima of the coordinate gradients and observed that the norm values remain finite and stable throughout training. The results can be found in Appendix E of our revised paper.

---

### Official Review · Reviewer_gFkG · 2025-11-01

**Soundness:** 3
**Presentation:** 3
**Contribution:** 3
**Rating:** 6
**Confidence:** 2

**Summary:**

The paper revisits sign-based stochastic optimization with momentum (a.k.a. Signum) and provides tighter convergence analyses. First, under separable smoothness and separable bounded noise, the authors show that signSGD with momentum achieves the classical non-convex rate $O(T^{-1/4})$ without the large-batch or unimodal-symmetric-noise assumptions used in earlier work. Under the standard $\ell_2$-smoothness and bounded-variance assumptions, they further prove an $O(d^{1/2}T^{-1/4})$ rate, improving the prior $O(dT^{-1/4})$ dependence for momentum-based sign methods. For distributed majority-vote settings, they propose an unbiased server-side sign operator and derive improved rates, e.g., $O(d^{1/2}T^{-1/2}+dn^{-1/2})$ and $O(\max(d^{1/4}T^{-1/4}, d^{1/10}T^{-1/5}))$, that outperform prior results in both $d$ and $T$. Empirically, CIFAR-10 (centralized) and CIFAR-100 (distributed, 8 nodes) experiments with 10 seeds support the claims.

**Strengths:**

1. The paper delivers a theoretical tightening for sign-based momentum methods in both centralized and distributed settings. On the centralized side, it attains the classical non-convex rate $O(T^{-1/4})$ for Signum under separable smoothness and bounded noise without resorting to large-batch or restrictive noise-shape assumptions, and under standard $\ell_2$-smoothness it improves the dimension dependence from $d$ to $d^{1/2}$. On the distributed side, introducing an unbiased server-side sign operator leads to sharper dependencies on $d,T$, and the number of workers $n$, strengthening the case for majority-vote style aggregation.

2. The technical pivot, controlling sign error through a cleaner estimator-error recurrence, reads as careful and broadly applicable, and the presentation is clear, with assumptions and algorithms spelled out and tables that position the results against prior analyses.

3. Empirically, though modest in scale, the CIFAR-10/100 studies with multiple seeds are consistent with the theoretical story and demonstrate that the proposed analyses map to competitive performance in practice.

**Weaknesses:**

1. The empirical scope is narrow: results focus on CIFAR-10 (centralized) and CIFAR-100 with eight nodes (distributed), leaving open how the methods behave in larger-scale, highly heterogeneous, or bandwidth-constrained regimes.

2. The theory relies on specific schedules for step size and momentum, yet the experiments use grid-tuned constants without ablations that test sensitivity to the prescribed schedules, which blurs the link between bounds and practice.

3. The comparative breadth could be stronger: beyond SGDM/AdamW and selected sign baselines, results omit contenders such as error-feedback compressors or recent variance-reduced sign methods that would sharpen the empirical positioning.

**Questions:**

See Weaknesses. In particular:
1. Can you extend the evaluation beyond CIFAR-10 (centralized) and CIFAR-100 with eight nodes (distributed), e.g., to larger-scale, in order to assess how the methods behave in those settings?

2. Can you add ablation testing sensitivity to the theorem-prescribed step-size and momentum schedules versus the grid-tuned constants, to clarify the link between the bounds and practice?

3. Could you include additional baselines, such as error-feedback compressors or recent variance-reduced sign methods?

---

> ### Author Response · Authors · 2025-11-20
>
> Thank you very much for your constructive comments!
>
> >W1 \& Q1: Can you extend the evaluation beyond CIFAR-10 and CIFAR-100 with eight nodes, e.g., to larger-scale, in order to assess how the methods behave in those settings?
>
> **Response**: We appreciate the reviewer’s suggestion to extend the evaluations to larger-scale tasks, and we agree that broader validation is valuable. In response to your suggestion, we have extended the evaluation to larger-scale datasets and models, specifically training the GPT-2 and Qwen3 models on the Alpaca dataset. The results can be found in Section 5.3, which demonstrate that our Signum method continues to outperform other methods such as signSGD, SGDM, AdamW, and EF-SignSGD (signSGD with error-feedback).
>
> >W2 \& Q2: Can you add ablation testing sensitivity to the theorem-prescribed step-size and momentum schedules versus the grid-tuned constants, to clarify the link between the bounds and practice?
>
> **Response**: We agree that an ablation study varying step sizes and momentum schedules would strengthen the numerical results.  In response, we have included such analyses in the revised manuscript.
> Specifically, we report the results with different momentum parameters and step sizes for the GPT-2 and Qwen3 training task in Appendix E. It can be seen that the results remain stable within certain ranges.
>
> >W3 \& Q3: The comparative breadth could be stronger: beyond SGDM/AdamW and selected sign baselines, results omit contenders such as error-feedback compressors or recent variance-reduced sign methods that would sharpen the empirical positioning. Could you include additional baselines, such as error-feedback compressors or recent variance-reduced sign methods?
>
> **Response**: Thank you for your suggestion! As requested, we have included the error-feedback method, i.e., EF-signSGD (Karimireddy et al., 2019), as the additional comparison algorithm in our experiments (see Section 5 of our revised paper). As can be seen, our Signum method outperforms the EF-signSGD methods in all tasks.

---

### Official Review · Reviewer_pMhc · 2025-11-01

**Soundness:** 2
**Presentation:** 2
**Contribution:** 1
**Rating:** 4
**Confidence:** 3

**Summary:**

This paper establishes theoretical convergence guarantees for a class of stochastic optimization algorithms that employ gradient-sign updates (signSGD) augmented with momentum.

The analysis primarily focuses on the **Signum** algorithm in centralized settings and its distributed variants based on majority voting.

The main motivation is to address limitations of prior analyses, which required either large batch sizes or restrictive noise assumptions to attain the optimal convergence rate of (O(T^{-1/4})).

Empirical results on image classification tasks (CIFAR-10/100) show fast convergence compared to several established baselines.

**Strengths:**

1. The paper shows that their algorithm reduces the large-batch requirements and improves the dimension dependency in the theory of sign-based optimization.

2. The analysis is conducted under multiple standard assumptions, making the results robust and widely applicable, and the distributed analysis also accounts for heterogeneous data settings.

3. The experimental results demonstrate superior performance in both centralized and distributed environments.

**Weaknesses:**

1. It remains unclear how the proposed algorithm compares with Ref. [1]. The method in Ref. [1] uses a fixed mini-batch size and imposes no noise assumptions, yet achieves an O(T^{-1/3}) complexity, whereas the present work reports only O(T^{-1/4}). Although the authors note that Ref. [1] assumes component-wise smoothness while this paper assumes global smoothness, both assumptions appear mild.

2. While the theory significantly improves the dependence on the dimension $d$, the experiments do not explicitly validate this. Performance gains are demonstrated on fixed-dimension tasks (e.g., CIFAR), but an ablation study varying $d$ would have provided stronger empirical support for the theoretical claims.

3. The related work section mentions error-feedback (Karimireddy et al., 2019) but does not discuss in depth how the proposed methods compare to these techniques in theory or practice, particularly regarding robustness and performance when the sign operation introduces significant bias.

Ref [1]. Wei Jiang, Sifan Yang, Wenhao Yang, and Lijun Zhang, Efficient Sign-Based Optimization: Accelerating Convergence via Variance Reduction, NeurIPS 37, pp. 33891–33932, 2024.

**Questions:**

We already have the Signum algorithm—why is there a need to introduce MVSM? Could you better illustrate the motivation behind MVSM?

---

> ### Author Response · Authors · 2025-11-20
>
> Thanks for your constructive comments! We have addressed the concerns you raised and revised our paper accordingly. We sincerely hope that the reviewer can examine them and reevaluate our paper.
>
> >W1: It remains unclear how the proposed algorithm compares with Ref. [1].
>
> **Response:** We would like to emphasize that Ref. [1] achieves the $O(T^{-1/3})$ rate under the **stronger average smoothness condition** (also referred to as mean-squared smoothness in some literature). This condition requires that each stochastic sample be smooth, i.e., $E[\Vert\nabla f(x;\xi_t)-\nabla f(y;\xi_t)\Vert^2]\leq L^2\Vert x-y\Vert^2$ for each $\xi_t$. In contrast, our $O(T^{-1/4})$ rate is derived under the standard smoothness assumption, which only requires that the objective function is smooth, i.e., $\Vert\nabla f(x)-\nabla f(y)\Vert\leq L \Vert x-y\Vert$. This difference is also reflected in the corresponding lower bounds for finding $\epsilon$-stationary points (Arjevani et al., 2023). Specifically, standard smoothness leads to a complexity lower bound of $O(\epsilon^{-4})$, while mean-squared smoothness yields a better lower bound of $O(\epsilon^{-3})$. Considering that the standard smoothness is more general and easier to satisfy, it is more meaningful to obtain guarantees for sign-based methods under such a milder assumption.
>
> Additionally, to achieve variance reduction in Ref. [1], their method requires computing two gradients $\nabla f(x_t;\xi_t)$ and $\nabla f(x_{t-1};\xi_t)$ at each step $t$. In contrast, we only require computing $\nabla f(x_t;\xi_t)$, avoiding the extra computational overhead. We have made these differences clearer in the related work section of the revised paper.
>
> >W2: Performance gains are demonstrated on fixed-dimension tasks (e.g., CIFAR), but an ablation study varying $d$ would have provided stronger empirical support for the theoretical claims.
>
> **Response**: We would like to clarify that the dimension $d$ is with respect to the optimization variable $x$, which represents the parameters of the learning model, rather than the input training data. Typically, a larger dimension $d$ corresponds to a larger learning model, and vice versa. In the original paper, we train the Resnet-18 and Resnet-50 models, whose dimensions are $d=12M$ and $d=25M$, respectively. We have also added more experimental results on much larger models and more complex tasks in the revised paper. Specifically, we train GPT-2 (with $d=125M$) and the popular Qwen3 (with $d=0.6B$) models on the Alpaca dataset. The results in Section 5.3 demonstrate that our method outperforms previous methods, including signSGD, SGDM, AdamW, and EF-SignSGD.
>
>
> >W3: The related work section mentions error-feedback (Karimireddy et al., 2019) but does not discuss in depth how the proposed methods compare to these techniques in theory or practice.
>
> **Response**: Thank you for this valuable suggestion! We would like to clarify that the error-feedback (Karimireddy et al., 2019) method differs from our approach in two key aspects: (1) The error-feedback update requires transmitting the information $(\Vert v_t \Vert_1/d) \text{sign} (v_t)$, while standard sign-based methods (including ours) only require transmitting $\text{sign}(v_t)$, which is more communication-efficient.
> (2) The error-feedback technique further assumes that the stochastic gradient is bounded, i.e., $E[\Vert\nabla f(x_t;\xi_t)\Vert_2^2]\leq\sigma^2$.
> As a result, our method obtains the convergence guarantees with less communication overhead and weaker assumptions compared to the error-feedback method. We have made these differences clearer in the related work section of our revised paper.
>
> Regarding practical performance, we have included the error-feedback method as one of the baselines in the newly added experiments involving the training of GPT-2 and Qwen3 models (see Section 5.3). The results show that our Signum method outperforms EF-SignSGD (Karimireddy et al., 2019) while maintaining lower communication overhead.
>
>
> >Q1: We already have the Signum algorithm—why is there a need to introduce MVSM? Could you better illustrate the motivation behind MVSM?
>
> **Response**: We would like to emphasize that the traditional Signum algorithm is designed for centralized settings, whereas our MVSM method is specifically introduced for distributed settings. Although previous literature (Bernstein et al., 2019) proposes **Signum with majority vote** to extend Signum to distributed settings, it can only provide convergence guarantees for simpler homogeneous settings and additionally supposes the assumption of unimodal symmetric noise. The primary motivation behind MVSM is to propose a method suited for challenging heterogeneous environments, achieving improved convergence rates compared to existing sign-based methods (Sun et al., 2023; Jin et al., 2021) without relying on additional strong assumptions. We have now clearly articulated this motivation at the beginning of Section 4.2.

---

### Author Response · Authors · 2025-11-20
**Global Response**

Dear Reviewers,

Thank you for the thoughtful and constructive feedback! Based on your comments, we have revised our manuscript (with changes highlighted in blue) and provided responses to address your concerns in detail. In the revised paper, we have added new experiments on the language model fine-tuning tasks in Section 5.3. Additionally, we introduced the error-feedback signSGD (EF-signSGD) as a new baseline to provide a better empirical evaluation. Furthermore, we have made the code publicly available via the following anonymous link: https://anonymous.4open.science/r/ICLR2026_Submission15147-DCFE, in line with Reviewer fqxZ’s request. We hope that these revisions and our detailed responses effectively address the concerns raised and enhance the overall quality of our submission.

Authors

---

### Author Response · Authors · 2025-12-01

Dear Area Chair,

We sincerely thank you and the reviewer for the careful assessment of our submission.  We believe we have addressed the concerns raised by the reviewers in our revised manuscript and rebuttal.

---

**For Reviewer pMhc**, we clarified that Ref. [1] relies on the stronger mean-squared smoothness, whereas our results hold under the standard smoothness assumption. Since standard smoothness is more general and employed by most stochastic optimization literature, analyzing sign-based methods under this assumption is more meaningful. We also resolved the misunderstanding regarding the dimension $d$, and added experiments on larger models and tasks (fine-tuning GPT-2 and Qwen3), including the error-feedback method as an extra baseline.

**For Reviewer fqxZ**, we provided the code as requested and specified how to implement $S_G$ in practice. We are pleased that the reviewer confirmed the concerns have been addressed and will raise the score.

Best regards,

The Authors

---

### Meta-Review · Area_Chair_Ejea · 2026-01-17

**Summary:**

This work focuses on sign-based stochastic algorithms for solving smooth and unconstrained optimization problems. The authors analyze a method where an averaging scheme is used at every iteration taking the convex combination of the previous state of the estimator and the new stochastic gradient sign. The authors get the same $T$ dependence in their convergence rate as Sun et al., 2023 (while improving the $d$ dependence) and they improve the $T$ dependence in the distributed setting.

It is worth contrasting this result with that of Jiang et al., 2024 that assumed the stronger mean-square smoothness on the function and obtained similar improvements (in fact, they could get an even better $T$ dependence due to the stronger assumption therein). I definitely appreciate the authors' claim that the assumption in Jiang et al., 2024 is stronger and hence the rate results are not comparable. However, this is indeed due to the difference of the "variance-reduction" schemes between Jiang et al., 2024 and this paper. It is well known that the averaging scheme in this submission does not require the mean-square smoothness assumption to reduce the variance (see for example "Stochastic Conditional Gradient Methods: From Convex Minimization to Submodular Maximization" by Mokhtari, Hassani, Karbasi, 2018). That is, the authors need to answer the following question: what is the difference between their analysis and Jiang et al., 2024 apart from removing the STORM-style variance reduction from Jiang et al., 2024 and inserting the averaging-based variance reduction. Indeed, both these estimators are classical. If the difference is only plugging in and out different estimators (which seems to be the case in my reading), I don't think this work would merit publication at a top venue. If there are further novelties in the analysis, the authors need to make it crystal clear in their next submission.

Another issue is the lack of precise parameters for the algorithms. Theorem 3 only gives the parameters in big-Oh notation and I also could not find the precise expression for the parameters. These are indeed necessary for the algorithm to be well-defined. The authors need to work out these parameters explicitly.

**Reviewer Concerns:**

The concerns of Reviewer pMhc about the comparison with Jiang et al., 2023 are only partly addressed. Even though I understand the difference of assumptions, hence the difference of rates, it is not clear if the differences in terms of analysis techniques are big enough. In particular, it does read like the difference between this submission and the previous work is changing the estimator (from STORM to averaging). This could be fine but the authors need to make this clear and explain the novelties in their analysis. Otherwise, only changing the estimator is not sufficient for acceptance.

The concern of Reviewer gFkG about the inconsistency between the required parameters for the theory and practice is not addressed.

**Reviewer Scores:**

I think that Reviewer pMhc would not increase their score since the author rebuttal does not clarify the technical novelties on top of Jiang et al., 2023, apart from changing the estimator (the latter issue is indeed my concern as well).

I think the scores of Reviewer gFkG and 2GAT would stay the same.

I think the Reviewer fqxZ would increase their score since their concerns are addressed.

---

### Decision · Program_Chairs · 2026-01-26

Reject